# STOCHASTIC SAMPLING FROM DETERMINISTIC FLOW MODELS

## ABSTRACT

Deterministic flow models, such as rectified flows, offer a general framework for learning a deterministic transport map between two distributions, realized as the vector field for an ordinary differential equation (ODE). However, they are sensitive to model estimation and discretization errors and do not permit different samples conditioned on an intermediate state, limiting their application. We present a general method to turn the underlying ODE of such flow models into a family of stochastic differential equations (SDEs) that have the same marginal distributions. This method permits us to derive families of *stochastic samplers*, for fixed (e.g., previously trained) *deterministic* flow models, that continuously span the spectrum of deterministic and stochastic sampling, given access to the flow field and the score function. Our method provides additional degrees of freedom that help alleviate the issues with the deterministic samplers and empirically outperforms them. We empirically demonstrate advantages of our method on a toy Gaussian setup and on the large scale ImageNet generation task. Further, our family of stochastic samplers provide an additional knob for controlling the diversity of generation, which we qualitatively demonstrate in our experiments.

## 1 INTRODUCTION

Deterministic flow models, including Rectified Flow (Liu et al., 2022), Flow Matching (Lipman et al., 2022; Tong et al., 2023), Stochastic Interpolants (Albergo & Vanden-Eijnden, 2022; Albergo et al., 2023), and probability flow ODE (Song et al., 2020) learn a reversible deterministic transport between two end distributions $p_0(x_0)$ and $p_1(x_1)$. Diffusion models require one of the distributions to be a Gaussian distribution, though generalizations exist (Yoon et al., 2024). In contrast, Rectified Flows, Stochastic Interpolants, and Flow Matching offer a general framework for learning deterministic transports, without this restriction. While deterministic transport enables efficient deterministic sampling, e.g. by the rectification procedure suggested by Liu et al. (2022), stochastic sampling may be desirable for: (1) robustness to estimation errors in the learned flow model, (2) ability to produce random samples conditioned on an intermediate state $x_t, t \in [0, 1]$, and (3) robustness to discretization error resulting from discrete step sampling from a continuous time model. We present a new theorem (Theorem 1) that provides a recipe to create an infinite family of parameterized stochastic samplers, given access to the flow field and the score function for the marginal distributions. Our result provides a general and unified view, while including a few existing proposals (e.g. in Huang et al. (2021); Berner et al. (2022)) as special cases.

The deterministic transport specifies a deterministic mapping between the samples from the two distributions and is realized as a learned vector field corresponding to an ordinary differential equation (ODE). However, if one distribution is chosen to be a Gaussian, these Flow models can be viewed as reparameterizations of other deterministic models that also choose a Gaussian as one of the distributions e.g. probability flow ODEs arising from Gaussian diffusion models. We refer to such models as *Gaussian flow* models. Transport map learning algorithms such as Gaussian flows are practical to train and enable applications like generative modeling (Ramesh et al., 2022; Lu et al., 2022; Saharia et al., 2022; Esser et al., 2024), stylization (Isola et al., 2017; Meng et al., 2022), and image restoration (Delbracio & Milanfar, 2023; Rombach et al., 2022; Lugmayr et al., 2022; Kawar et al., 2022), to name a few. However, corresponding deterministic sampler has limitations that we empricially demonstrate on a toy Gaussian task, where it exhibits a bias and consistently underestimates the variance of the target distribution, as seen in Figure 2. To enable stochastic

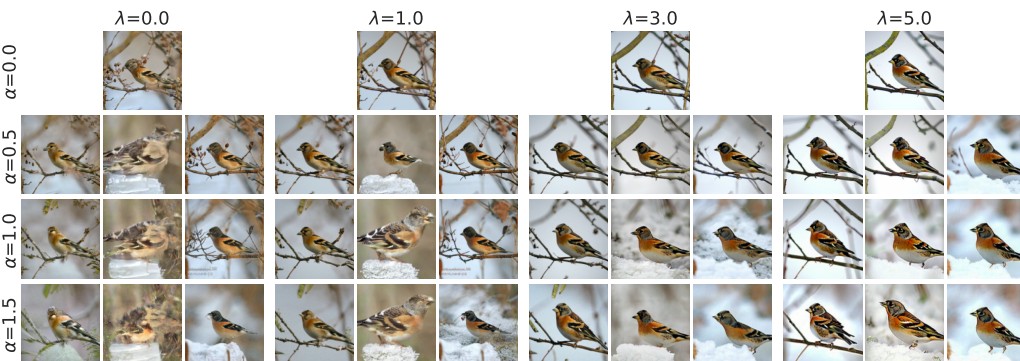

Figure 1: **Stochastic sampling improves diversity at all classifier-free guidance levels.** We visualize samples from a rectified flow model at four classifier-free guidance levels $\lambda$ (Section 3.3) and at four stochasticity scales $\alpha$ for NonSingular (Table 1). Three samples are shown for each configuration where the sampling starts at the same draw from $p_1(x_1)$. When $\alpha = 0$, the sampler is deterministic and samples are the same (therefore we show only one). When $\lambda = 0$, there is no classifier-free guidance. Note the increased diversity as $\alpha$ increases. More examples in Figure 12.

sampling from such deterministic models, we provide a special case of our general result to turn the underlying ODE of Gaussian flow models into a family of stochastic differential equations (SDEs) that have the same marginal distributions. Our stochastic samplers allow trading the bias of deterministic sampler for increased variance in the estimated mean and variance parameters (Figure 4). Since, our method requires access to the score function for the marginal distributions, we impute it directly from the given flow model, alleviating the need for learning it separately. This method permits us to derive families of *stochastic samplers*, for fixed (e.g., previously trained) *deterministic* Gaussian flow models, that allow flexible and time dependent injection of stochasticity during sampling, enabling both deterministic and stochastic sampling. This additional degree of freedom allows exploration of stochastic samplers that can help alleviate the issues with the deterministic samplers and outperform them. We demonstrate this empirically on a toy Gaussian setup, as well as on the large scale ImageNet generation task. The stochastic samplers also provide an additional knob for controlling the diversity of generation as we qualitatively demonstrate in our experiments, and are compatible with classifier-free guidance (Ho & Salimans, 2022), as can be seen in Figures 1 and 12.

**Our key contributions are: (1)** Specification of a flexible family of SDEs (Theorem 1) that have the same marginal distributions as a given SDE or a flow model, enabling exploration of sampling schemes for a given fixed model, **(2)** Derivation of new as well as existing special cases directly from Theorem 1 (Corollary 1.1 and Corollary 1.2) demonstrating generality of Theorem 1, **(3)** Study of a set of SDE families corresponding to Gaussian flow models, derived using Theorem 1, on both a toy as well as a large scale ImageNet setup, demonstrating flexible stochastic sampling and controllable diversity in generation, *without requiring retraining* (Table 1, Figures 1 and 12).

## 2 BACKGROUND

**Notation.** Throughout this work we use small Latin letters $t, x, y$ etc. to represent scalar and vector variables, $f, g$ etc. to represent functions, Greek letters $\alpha, \beta$ etc. to represent (hyper-)parameters, and capital letters $G$ to represent matrices. With a slight abuse of notation we use lower case letters $x$ to represent both the random variable and a particular value $x \sim p(x)$. Whenever unambiguous, we suppress the dependence of state $x_t$ on time $t$ as $x \equiv x_t$, and dependence of functions on state $x_t$ and time $t$ as $f \equiv f(x_t, t)$ to simplify notation.

We briefly discuss rectified flow and continuous time diffusion models. Refer to Liu et al. (2022); Song et al. (2020) for details.

### 2.1 RECTIFIED FLOW

Let $x_0 \sim p_0(x_0) \in \mathbb{R}^d$ be the draws from the data distribution $p_0$ that we are interested in learning and sampling from. Let $x_1 \sim p_1(x_1) \in \mathbb{R}^d$ be an easy to sample source distribution. Loosely, the key

idea behind the diffusion and flow family of models is to learn a mapping that either stochastically or deterministically transforms a sample from $p_1$, in an iterative manner, to produce a sample from $p_0$. Let $\nu(x_0, x_1)$ be an arbitrary coupling distribution for the two random variables $x_0, x_1$ such that $p_0(x_0) = \int \nu(x_0, x_1) dx_1, p_1(x_1) = \int \nu(x_0, x_1) dx_0$. A simple choice is the product of the two: $\nu(x_0, x_1) \equiv p_0(x_0)p_1(x_1)$. To construct a rectified flow first an interpolation between the two variables is defined as $x_t \equiv h(x_0, x_1, t)$ that is differentiable w.r.t. time. The default interpolation proposed and studied in Liu et al. (2022) is:

$$x_t = (1-t)x_0 + tx_1, \quad t \in [0, 1]. \tag{1}$$

With the above, rectified flow learns a vector field $v(x_t, t)$ by minimizing the following objective:

$$v(x, t) = \arg\min_{v'} \mathbb{E}_{(x_0, x_1) \sim \nu} \left[ \int_0^1 \left\| \frac{dx_t}{dt} - v'(x_t, t) \right\|^2 dt \right]. \tag{2}$$

The solution to the above optimization problem is $v(x, t) \equiv \mathbb{E}[x_1 - x_0 | x, t]$ and is referred to as 1-Rectified flow. Since $v(x, t)$ is not available in closed-form in general, $v$ is typically parameterized with parameters $\theta$ and optimization in Equation (2) is performed w.r.t. $\theta$. In the rest of the paper, we drop this dependence on the parameters in notation as we assume a model $v(x, t)$ to be given. Note that a closed-form expression is available when $p_0, p_1$ are Gaussian (see Appendix F). We use this expression for the toy setup in our experiments. For example, the biased deterministic sampler in Figure 2 is using the ground truth flow field. Once the flow $v(x_t, t)$ is estimated, samples from $p_0(x_0)$ can be produced by drawing a sample from $p_1(x_1)$ and simulating the flow backward in time, using:

$$dx = v(x, t)dt \tag{3}$$

## 2.2 Score based diffusion with stochastic differential equations

The general idea in this family of methods is to specify a forward stochastic process that slowly transforms the data density $p_0(x_0)$ into an easy to sample source density $p_1(x_1)$. Song et al. (2020) specified such a process using an Itô SDE of the following form:

$$dx = f(x, t)dt + G(x, t)dW_t \tag{4}$$

where $f(x, t) : \mathbb{R}^d \times [0, 1] \to \mathbb{R}^d$ is the drift coefficient, $G(x, t) : \mathbb{R}^d \times [0, 1] \to \mathbb{R}^d \times \mathbb{R}^d$ is state and time dependent diffusion coefficient and $W_t$ is the Wiener process. Choosing[1] $G \equiv g(t) : [0, 1] \to \mathbb{R}$ and using results from Anderson (1982), a reverse time SDE can be specified that has the same marginals as Equation (4):

$$dx = [f(x, t) - g^2(t) \nabla_x \ln p_t(x)]dt + g(t)d\tilde{W}_t \tag{5}$$

where $\tilde{W}_t$ is a standard Wiener process with time running backwards. Note that the time reversal requires access to the score function $\nabla_x \ln p_t(x)$. Score matching (Vincent, 2011) can be used to learn an estimate for the score for all $t$ (Song et al., 2020), which can then be used to simulate reverse time dynamics starting with a sample from $p_1(x_1)$ to produce a sample from $p_0(x_0)$ at $t = 0$. A forward deterministic process can also be derived from the above that has the same marginal densities $p_t(x)$:

$$dx = \left[ f(x, t) - \frac{1}{2} g^2(t) \nabla_x \ln p_t(x) \right] dt \tag{6}$$

The above ODE is also referred to as the probability flow ODE. Samples can be generated using the above ODE in a similar fashion as rectified flow, by simulating the ODE backwards in time.

## 3 Deriving stochastic samplers

**Method intuition.** Probability flow ODEs (Song et al., 2020), proposed in the context of diffusion models, provide a deterministic sampling method for diffusion models. These ODEs have the same marginal distribution $p_t(x)$ at all $t$ as the original SDE from which they are derived. Here, we take the reverse path: we start from an ODE (corresponding to the Gaussian flow model) and deduce the family of SDEs that have the same marginal distributions at all $t$ as the original ODE. Before we introduce the general result, we will show a naive approach that gives an SDE with a problematic singularity, motivating the need for the generalization.

---

[1]Song et al. (2020) provide general results for $G(x, t)$ which we omit here for brevity.

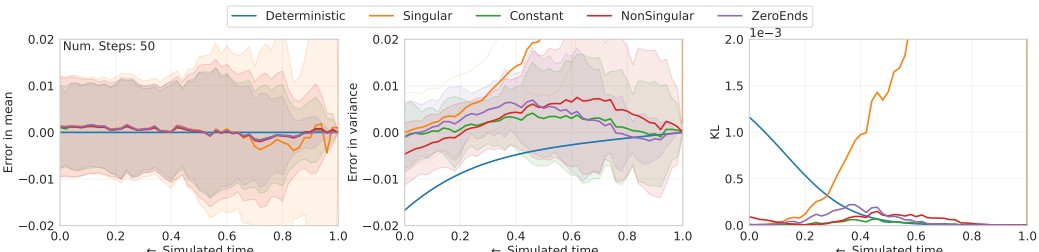

Figure 2: **Discretization of deterministic flow leads to bias.** Comparison of samplers from Table 1 on the two Gaussian toy problem (Appendix G). Deterministic underestimates the variance parameter, but the stochastic samplers avoid that issue, in exchange for variance in the parameter estimation. Singular's variance diverges if we start from $t = 1$, so instead we start the sampler at $t = 1 - 10^{-3}$, which allows it to eventually converge by $t = 0$.

### 3.1 A SINGULAR SDE CORRESPONDING TO GAUSSIAN FLOW

For Gaussian flow, $p_1(x_1) \equiv N(x_1; \mu_1, \sigma_1^2 I)$ is assumed to be Gaussian. With interpolation $x_t = (1 - t)x_0 + tx_1$, the perturbation kernel $p(x_t|x_0) = N(x_t; (1 - t)x_0 + t\mu_1, t^2\sigma_1^2 I)$ is also Gaussian.

Note that since $x_0, x_1$ are independent, we can directly infer the first and second moments $\mu_t, \Sigma_t$ for the marginals $p_t(x)$ as $\mu_t = (1 - t)\mu_0 + t\mu_1$ and $\Sigma_t = (1 - t)^2\Sigma_0 + t^2\sigma_1^2 I$. With these constraints and choosing $\mu_1 \equiv 0, \sigma_1 \equiv 1$, we can solve for drift and diffusion coefficients that yield the same marginal distributions:

$$f(x,t) = -\frac{x}{1 - t} \qquad\qquad g(t) = \sqrt{\frac{2t}{1 - t}} \qquad (7)$$

See Appendix A for the details and a more general expression for arbitrary $\mu_1, \sigma_1$. The coefficients $f(x,t), g(t)$ are singular at the boundary $t = 1$ of the interval. Consequently, simulation methods such as Euler-Maruyama, that need $f(x,t), g(t)$ to be Lipschitz are not guaranteed to work at the boundary (see Figure 2 and Section 4.1). We refer to this SDE as the Singular SDE. An empirical trick that is often used in such cases is to assume $p_{1-\epsilon}(x_{1-\epsilon}) \approx p_1(x_1), \epsilon \ll 1$. However, this can lead to unpredictable behavior and we show how to avoid it in the following section.

### 3.2 SET OF SDES THAT SHARE THE SAME MARGINAL DISTRIBUTION $p_t(x)$

First we state our general result with the diffusion coefficient $G(x,t)$ a function of both the state $x$ and time $t$, and then state simpler forms more directly applicable to models used in practice.

**Theorem 1.** *Let $p_t(x)$ be the probability density of the solutions of the SDE in Equation (4) evolving as $\frac{\partial p_t}{\partial t}$. Then, the probability density of solutions of the following set of SDEs, indexed by $\tilde{G}, \gamma_t$, also evolves as $\frac{\partial p_t}{\partial t}$.*

$$dx = \bar{f}(x,t)dt + \bar{G}(x,t)dW_t \qquad (8)$$

*where*

$$\bar{f} = f - \frac{1}{2}\left(\nabla \cdot [(1 - \gamma_t)GG^T - \tilde{G}\tilde{G}^T] + [(1 - \gamma_t)GG^T - \tilde{G}\tilde{G}^T] \cdot \nabla \ln p_t\right) \qquad (9)$$

$$\bar{G} = [\gamma_t GG^T + \tilde{G}\tilde{G}^T]^{\frac{1}{2}} \qquad (10)$$

*and $\tilde{G} \equiv \tilde{G}(x,t), \gamma_t \geq 0$ are arbitrary functions such that the solutions of Equation (8) exist and are unique.*

Proof of Theorem 1 is given in Appendix C and follows from manipulations of Fokker-Planck-Kolmogorov (FPK) equations corresponding to Equation (8).

Theorem 1 implies that given the same initial density $p_0(x)$, evolution according to both Equation (4) and Equation (8) will have the same marginal densities $p_t(x)$ for all times $t$. Further, Equation (8) can

be simulated backward in time using the result from Anderson (1982), again with the same marginal densities $p_t(x)$. Consequently, Equation (4) can be simulated forward or backward in time using any member of the family specified by Equation (8). Note that setting $\gamma_t = 1, \tilde{G} = 0$ recovers the original SDE in Equation (4), while setting $\gamma_t = 0, \tilde{G} = 0$ recovers the general probability flow ODE from Song et al. (2020, eq. 37). Additionally, $\tilde{G}$ is particularly useful for deterministic flow models, further discussed in Corollary 1.2. Theorem 1 gives a recipe for developing particular samplers, such as those in the remainder of this section, some of which have appeared in the literature. A priori, Theorem 1 cannot determine which concrete sampler will be best for a given application, but since the samplers do not require any training to use, it is possible to postpone the choice of sampler to an empirical analysis at test time.

The flexibility afforded by Equation (8) is particularly useful **(1)** in the presence of singularities in the drift and diffusion coefficients $f$ and $G$ respectively of Equation (4), **(2)** in the presence of errors resulting from finite discretization, and **(3)** for flexible specification of the diffusion coefficient in generative applications. Our experimental evaluations primarily focus on these aspects of Theorem 1.

A direct consequence of Theorem 1, by defining $\tilde{G} \equiv 0, G \equiv g(t)I$, is the following corollary applicable to commonly used generative diffusion models with additive noise:

**Corollary 1.1.** *For the SDE in Equation* (4) *with $G \equiv g(t)I$, a subset of SDEs prescribed by Theorem 1 and indexed by $\gamma_t$ is:*

$$dx = \left[ f(x,t) - \frac{(1-\gamma(t))g^2(t)}{2} \nabla_x \ln p_t(x) \right] dt + \sqrt{\gamma(t)} g(t) dW_t \tag{11}$$

Proof in Appendix D. Note that choosing $\gamma_t = 0$ results in the probability flow ODE specified in Equation (6). Intuitively, the members in the family differ in terms of the amount of noise injected as a function of time. $\gamma_t = 0$ yields a purely deterministic simulation; $\gamma_t > 0$ yields a variety of stochastic simulations. Further, similar special cases discussed in Huang et al. (2021) and Berner et al. (2022) also directly follow from Theorem 1 as well.

Some properties of Corollary 1.1:

1. $\gamma(t)$ can be chosen at sampling time and doesn't affect the training of the score function.

2. With $\gamma(t) = \hat{\gamma}^2(t)g^{-2}(t)$, where $\hat{\gamma}(t)$ is an arbitrary function (satisfying constraints of Theorem 1), we can choose an arbitrary diffusion term at sampling time. For example, choosing $\gamma_t = \gamma^2/g^2(t)$ leads to a constant diffusion coefficient.

3. For the SDE specified by Equation (7), we can choose $\gamma(t) = (1-t)\hat{\gamma}^2(t)g^{-2}(t)$ to get rid of the singularity in the diffusion term.

Note that Theorem 1 can be used whenever we have access to the score function $\nabla_x \ln p_t$. Next, we first construct a specialized solution based on Theorem 1 for deterministic flow models that enables flexible control of both drift and diffusion coefficients, and apply it to the special case of deterministic Gaussian flows where the score function can be imputed from the velocity field (Section 3.3). Recall that deterministic flows specify a transport via the ODE $dx = v(x,t)dt$. This ODE can be viewed as an SDE where the diffusion term has been set to zero. Choosing $G \equiv 0, \tilde{G} \equiv \tilde{g}(t)I$ in Theorem 1 gives Corollary 1.2, which enables deriving stochastic samplers for Gaussian flow models:

**Corollary 1.2.** *For the ODE in Equation* (3)*, a subset of SDEs prescribed by Theorem 1 and indexed by $\tilde{g}(t)$ is*

$$dx = \left[ v(x,t) + \frac{\tilde{g}^2(t)}{2} \nabla_x \ln p_t(x) \right] dt + \tilde{g}(t) dW_t \tag{12}$$

Proof in Appendix E. Corollary 1.2 enables flexible specification of a time dependent diffusion coefficient $\tilde{g}(t)$, allowing the introduction of stochasticity in the simulation of otherwise deterministic models, *purely at sampling time*. Note that Equation (12) requires access to the score function $\nabla_x \ln p_t(x)$ for the marginal distributions $p_t(x)$. In Section 3.3, we describe how the score function can be imputed from the learned flow model $v(x,t)$ for the special case of Gaussian flow models. It can be verified that the particular choice of $f$ and $g$ in Equation (7) satisfy Equation (12) by using the expression for the score from Equation (13).

Table 1: Examples of SDEs that have the same marginal distribution $p_t(x)$ as a given Gaussian flow specified by $v \equiv v(x,t)$. $\alpha \geq 0$ is a scale parameter that varies the magnitude of the diffusion coefficient $g$. Each of these behaves differently when discretized and simulated (Figure 2 and Appendix J.2). These and infinitely many more can be constructed using the scheme in Equation (12).

| Name | $\tilde{g}(t)$ | $\tilde{f}(x,t)$ | Description |
|---|---|---|---|
| Deterministic | $0$ | $v$ | Base flow model |
| Constant | $\alpha$ | $v + \frac{\alpha^2}{2}\nabla_x \ln p_t$ | Constant $g$, singular $f$ |
| Singular | $\alpha\sqrt{t/(1-t)}$ | $v + \frac{\alpha^2}{2}\frac{t}{1-t}\nabla_x \ln p_t$ | Singular $g, f$ |
| NonSingular | $\alpha\sqrt{t}$, | $v + \frac{\alpha^2}{2}t\nabla_x \ln p_t$ | Non-singular $g, f$ |
| ZeroEnds | $\alpha\sqrt{t(1-t)}$, | $v + \frac{\alpha^2}{2}t(1-t)\nabla_x \ln p_t$ | Non-singular $g, f, g(0) = g(1) = 0$ |

While infinitely many choices are available for $\tilde{g}$, we consider a few interesting ones listed in the Table 1, constructed by choosing integer powers of $t$ and $1 - t$ and introducing a scaling coefficient $\alpha$, for experimental evaluations. Note that the only degree of freedom in Table 1 is the choice of $\tilde{g}(t)$, which determines $\tilde{f}(x,t)$, given the flow field $v(x,t)$ and the score $\nabla_x \ln p_t(x)$. The $f(x,t)$ is singular in Constant because the score $\nabla_x \ln p_t(x_t)$, as computed in Equation (13), has $t$ in the denominator, making $f(x,t)$ singular at $t = 0$. The choice in NonSingular precisely eliminates this singularity. Figure 2 compares these choices in a toy setup; Section 4 has comparisons on ImageNet.

### 3.3  SCORE FUNCTION AND CLASSIFIER FREE GUIDANCE FOR A GAUSSIAN FLOW MODEL

Recall that Theorem 1 requires access to the score function. For Gaussian flows, the score function can be inferred from the velocity field itself, alleviating the need to learn it separately. This result is known (see e.g. Zheng et al. (2023) in the context of flow matching) and we present it here in our setting. For Gaussian flows, with $p_1(x_1) \equiv N(x_1; \mu_1, \sigma_1^2 I)$ and interpolation specified in Equation (1), the score can be computed as:

$$\nabla_x \ln p_t(x) = \frac{-(1-t)v(x,t) + \mu_1 - x}{t\sigma_1^2} \tag{13}$$

where $v(x,t) = \mathbb{E}[x_1 - x_0 | x, t]$ is the estimated flow. Proof is provided in Appendix B. Note that the score function can also be estimated given $\mathbb{E}[x_0 | x, t]$ or $\mathbb{E}[x_1 | x, t]$. In summary, the expression follows directly from using results from Denoising Score Matching (Vincent, 2011) and the Gaussianity of $p_1(x_1)$. Similar expressions can be derived for other interpolations that are linear in $x_0, x_1$. With access to the score function and linearity of Equation (13) in $v$ we can define classifier free guided (Ho & Salimans, 2022) Gaussian flow as:

$$v_{\text{cfg}}(x,t,c) = (1 + \lambda)v(x,t,c) - \lambda(v(x,t,c = \varnothing) \tag{14}$$

where $c$ indicates extra conditioning information as in classifier free guidance, $\varnothing$ indicates no conditioning and $\lambda$ specifies the relative strength of the guidance. $\lambda = 0$ reduces to class conditional sampling, while $\lambda > 0$ puts a larger weight on the conditioning, biasing the sample towards the modes of the conditional distribution. Using classifier-free guidance with a stochastic sampler will, of course, give diversity that isn't possible with a deterministic sampler, as can be seen in Figure 1. Note that Xie et al. (2024); Dao et al. (2023); Zheng et al. (2023) also consider related definitions in the context of flow matching.

## 4  EXPERIMENTS

Our method allows us to identify a family of SDEs that correspond to a given deterministic Gaussian flow model, enabling construction of stochastic samplers with flexible diffusion coefficients. In our experiments we compare various samplers derived from the corresponding SDEs in Table 1, using Euler-Maruyama, for a given Gaussian flow model *without any additional training*.

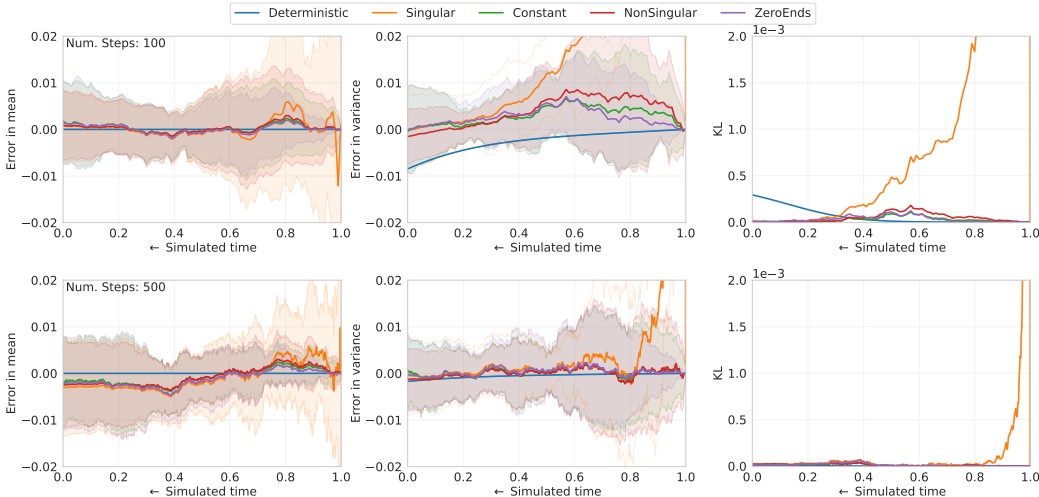

Figure 3: **Stochasticity is most helpful at coarser discretizations.** We visualize the effect of coarseness of discretization by sampling for 100 and 500 sampling steps. See Figure 2 for the same plots at 50 steps, which shows more extreme bias in variance for Deterministic and Singular.

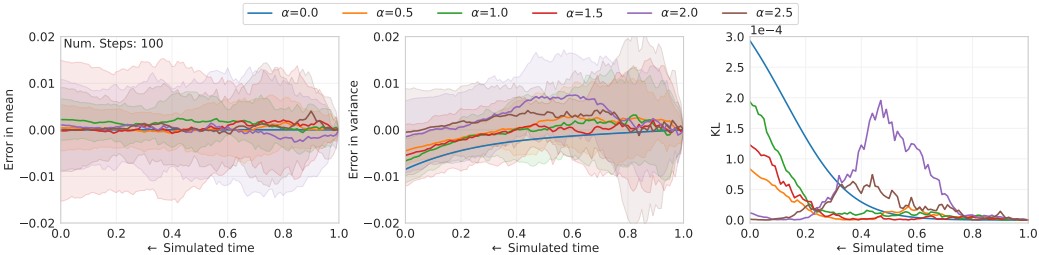

Figure 4: **Stochasticity helps mitigate bias.** We plot the error in mean and error in variance for NonSingular for a set of diffusion coefficient scales $\alpha \in \{0.0, 0.5, 1.0, 1.5, 2.0, 2.5\}$. Estimates for variance at $t = 0$ improve as $\alpha$ increases, leading to a drop in KL divergence from the true distribution. However, with very high $\alpha$ values intermediate marginals develop a bias.

### 4.1 COMPARISON ON ESTIMATING MARGINAL STATISTICS FOR A TWO GAUSSIAN TOY PROBLEM

We start by considering a toy problem where both $p_0$ and $p_1$ are Gaussian. See Appendix G for details of the experimental setup and Appendix I for a JAX (Bradbury et al., 2018) implementation of NonSingular.

**Discretization of deterministic flow leads to bias.** In Figure 2, with 50 sampling steps, we observe that the estimate for the mean is fairly accurate for all samplers for the entirety of the interval $t \in [0, 1]$. However, the samplers differ in their behavior for variance. Deterministic exhibits a noticeable bias and underestimates the variance (with zero variance in its estimate), with the worst estimate at $t = 0$. Stochastic samplers provide noticeably better estimates at $t = 0$, but with increased variance.

**Stochasticity is most helpful at coarser discretizations.** In Figure 3 we study the effect of the number of discretization steps on the different samplers (also see Figure 2 for 50 steps). While mean estimates are accurate for all methods, Deterministic gets increasingly biased for variance estimates as the number of sampling steps is decreased. Stochastic samplers perform consistently well at various discretization levels for $t = 0$, with significantly better estimates for fewer sampling steps. Note that Singular has very large bias as well as variance closer to $t = 1$; those improve with finer discretization. Since Constant also has a singularity, but only in the drift term $f$, we conclude that the instability is primarily due to the singularity in Singular's diffusion term.

Table 2: **Stochasticity can improve FID.** Comparison of various samplers at their best $\alpha$ values with 300 sampling steps for ImageNet image generation task at two resolutions.

| Sampler | $64 \times 64$ | | $128 \times 128$ | |
|---|---|---|---|---|
| | FID | $\alpha$ | FID | $\alpha$ |
| Deterministic | $3.07 \pm 0.01$ | 0.0 | $5.19 \pm 0.02$ | 0.0 |
| Singular | $3.07 \pm 0.01$ | 0.08 | $5.13 \pm 0.04$ | 0.14 |
| Constant $g$ | $2.97 \pm 0.04$ | 0.08 | $5.17 \pm 0.05$ | 0.1 |
| NonSingular | $\mathbf{2.95 \pm 0.01}$ | 0.56 | $\mathbf{4.93 \pm 0.06}$ | 0.42 |
| ZeroEnds | $\mathbf{2.95 \pm 0.01}$ | 0.54 | $5.03 \pm 0.01$ | 0.52 |

**Stochasticity helps mitigate bias.**    In Figure 4 we study the effect of diffusion coefficient scale $\alpha$ on the NonSingular sampler at 100 sampling steps. Finite discretization introduces a bias in the deterministic sampler (when $\alpha = 0$), where the variance is consistently underestimated and is worst at $t = 0$. Increased stochasticity with increasing diffusion coefficient scale ($\alpha > 0$) helps mitigate this bias at the cost of increased variance. This can be seen in the figure with larger $\alpha$ values yielding better estimate of the variance, although with larger variance in the estimate.

## 4.2    COMPARISON OF SDEs FOR RECTIFIED FLOWS ON IMAGENET GENERATION

We compare the behavior of various SDEs on the sampling quality for large scale image generation using the ImageNet (2012) dataset (Deng et al., 2009; Russakovsky et al., 2015). We train rectified flow models at two different image resolutions ($64 \times 64$ and $128 \times 128$) and compare their sample quality using the Frechet Inception Distance (FID) metric (Heusel et al., 2017) for class conditional samples. See Appendix H for setup details. The results show that small but statistically significant differences exist between samplers even for metrics like FID, but the optimal sampler is likely to be application and model specific.

**Stochasticity can improve FID.**    In Table 2 we report the best FID using each SDE in Table 1 for two image resolutions using 300 sampling steps, along with the corresponding diffusion term scale $\alpha$ and one standard deviation confidence interval. Two key observations stand out: **(1)** stochastic samplers tend to produce better FIDs, and **(2)** the two non-singular samplers have much better FIDs than Deterministic or Singular. Note that observation **(1)** has also been made previously for probability flow ODEs (Song et al., 2020). The addition of a parameter $\alpha$ to control the strength of the stochasticity while keeping the marginal distribution $p_t$ unchanged (Theorem 1), permits principled post-training optimization of the metrics like FID.

**Non-singular samplers work well over a broad range of $\alpha$.**    In Figures 5 and 7 we show how the FID varies with $\alpha$ for each sampler for two different image resolution models. NonSingular and ZeroEnds attain better FID in general and are better behaved over a much larger range of the diffusion coefficient scale $\alpha$ at both resolutions. These samplers both have small diffusion coefficients $g(t)$ close to $t = 0$; their performance indicates that noise near $t = 0$ is particularly harmful. The low variance of ZeroEnds in comparison to NonSingular indicates that a large diffusion coefficient near $t = 1$ tends to introduce variance in the final FID.

**Stochasticity makes FID robust to discretization.**    In Figure 6 we compare the effect of the number of sampling steps on FID for various samplers at two image resolutions. We set $\alpha^2$ proportional to the number of sampling steps with the maximum value provided by Table 2. Again the non-singular samplers perform better than Deterministic at all discretization levels.

**Stochastic sampling improves diversity at all classifier-free guidance levels.**    In Figures 1 and 12 we show samples from NonSingular using classifier-free guidance (Section 3.3), varying both $\alpha$ and $\lambda$, the guidance weight. In all cases, we can see that diversity increases with $\alpha$, and class typicality increases with $\lambda$.

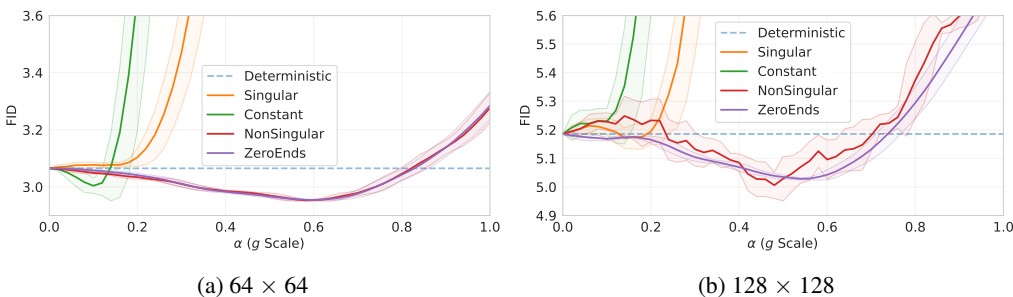

(a) $64 \times 64$        (b) $128 \times 128$

Figure 5: **Non-singular samplers work well over a broad range of $\alpha$.** Plots of FID for each sampler as the diffusion coefficient scale $\alpha$ is increased. Note that at $\alpha = 0$ all samplers coincide. See Figure 7 for a larger range of FIDs.

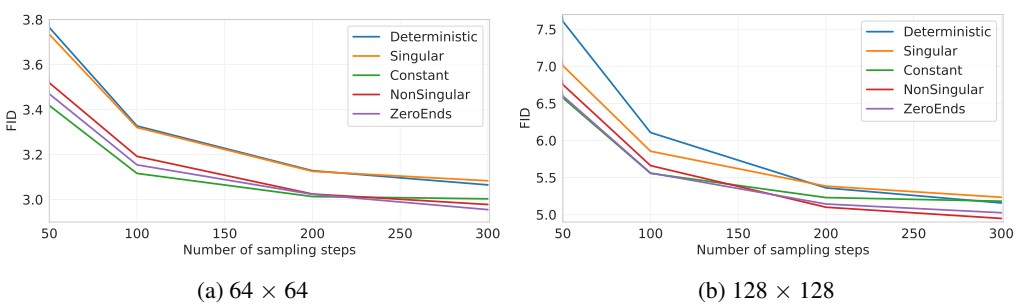

(a) $64 \times 64$        (b) $128 \times 128$

Figure 6: **Stochasticity makes FID robust to discretization.** We compare the effect of number of sampling steps on FID. Deterministic is always worse than the non-singular samplers.

**Qualitative comparisons.** For qualitative comparisons, we visualize a few samples at various diffusion coefficient scales using different SDEs in Figures 9 to 11. All samples in a column are generated by starting at the same draw $x_1 \sim p_1(x_1)$; different columns start from different draws. Noise scale $\alpha$ gets progressively larger as we move down the rows. For Constant, we observe that samples get increasingly noisy with increasing $\alpha$ indicating accumulating errors with increasing scale. The samples from NonSingular look better, as expected from Figure 5. Lastly, samples from Singular change much more rapidly in comparison to the other samplers, indicating that the singularities in the SDE coefficients increase the effect of noise.

## 5 RELATED WORK

Transport learning methods learn a mapping between two distributions, where the learned model can transform a sample from one distribution into a sample from the other one. Typically, one of the distributions is easy to sample (such as a Gaussian) and the other one is the data distribution that one is interested in modeling. The learned mapping can either be deterministic or stochastic. A thorough overview of related areas can be found in Yang et al. (2024).

**Deterministic transport.** Deterministic transport methods implement a change of variable, either explicitly or approximately, that can be used to uniquely map a sample from one distribution to the other. The normalizing flow family (Rezende & Mohamed, 2015; Dinh et al., 2017; Kingma & Dhariwal, 2018) of methods construct an explicit invertible model that realizes this map either in one step or a finite number of discrete steps. Neural ODEs (Chen et al., 2018; Grathwohl et al., 2019) generalize from discrete steps to a continuous time mapping by inferring and learning the gradient field for all times. However, Neural ODEs are difficult to train due to the need for simulating the ODEs as part of the training. Rectified flows, flow matching, and iterative denoising methods (Liu et al., 2022; Lipman et al., 2022; Tong et al., 2023; Heitz et al., 2023; Delbracio & Milanfar, 2023) either implicitly or explicitly specify a continuous mapping and learn a model to approximate the continuous time mapping. Similarly, probability flow ODEs (Song et al., 2020) learned by diffusion models (Sohl-Dickstein et al., 2015) approximate an implicitly defined continuous mapping. Our

work is useful for flexible sampling from such pre-trained continuous time deterministic Gaussian flows, or more generally where the score function for all the marginal distributions is either provided or can be deduced from the learned flow model.

**Stochastic transport.** Stochastic transport methods learn a stochastic mapping, where a sample from one distribution gets stochastically mapped to a sample from the other. Gaussian diffusion models are a salient example of such discrete Sohl-Dickstein et al. (2015); Ho et al. (2020) or continuous time Song et al. (2020); Kingma et al. (2021) mappings where one of the distributions is constrained to be Gaussian. Several generalizations have been proposed that extend from Gaussian to more general families of distributions Yoon et al. (2024). The stochastic interpolants framework (Albergo & Vanden-Eijnden, 2022; Albergo et al., 2023; Ma et al., 2024) further generalizes to a larger family of distributions by introducing a random latent variable allowing efficient estimation of the score function at all times. Our work is directly applicable to models learned with such methods where the score function is accessible and can be used to construct and explore a large family of samplers. The convergence rates of diffusion models have been studied in Chen et al. (2023); Benton et al. (2023) with respect to the number of data samples and dimensionality. However, since our method does not require retraining, it does not affect these properties of the original training algorithms.

**Schrödinger bridge and optimal transport.** These methods consider a more general problem of learning transport maps with additional constraints. k-Rectified flows Liu et al. (2022) provide an iterative procedure for tackling deterministic optimal transports for a family of costs, while the more general Schrödinger bridge problem, viewed as an entropy regularized optimal transport, is an active area of research Shi et al. (2024); Liu et al. (2023). Our work is complementary to these methods as we focus on flexible sampling, given the access to the score function for the marginal distributions.

## 6 CONCLUSION

We introduced a general method to identify a family of SDEs that have the same marginal distribution as a particular SDE, including the special case where the diffusion coefficient of the given SDE is zero. This special case corresponds to flow models which naively only support deterministic sampling. Our method enables flexible construction of stochastic samplers for such deterministic models where the diffusion coefficient can be chosen at sampling time from an infinitely large set of possibilities in an application and evaluation metric dependent way. Our method requires explicit access to the score function, in absence of which it is limited to a subset of flow models where the score function can be derived from the given flow model. However, this set includes currently popular rectified flow and diffusion models where one of the distributions is Gaussian.

## ETHICS STATEMENT

As a general technique for improving sampling from flow models at inference time, this work has minimal ethical implications beyond those common to most machine learning research. It is possible that malevolent actors could generate more convincing samples from existing models using this work, but it does not provide a fundamentally new capability to an attacker, so we consider the ethical risk to be low.

## REPRODUCIBILITY STATEMENT

We provide proof of Theorem 1, Corollary 1.1, and Corollary 1.2 in Appendix C, Appendix D, and Appendix E, respectively. We provide example code for one of our samplers in Figure 8; others are straightforward to reproduce using it as an example and following Table 1. $\alpha$ is the main hyperparameter of interest; we specify it in Table 2 for each sampler on the large scale ImageNet experiment.

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

## A    DERIVATION OF SINGULAR SDE

We consider the following SDE with additive noise; i.e., the diffusion coefficient $g$ is only a function of time.

$$dx = f(x, t)dt + g(t)dW_t \tag{15}$$

The perturbation kernel $p(x_t|x_0)$ corresponding to rectified flow is Gaussian, with $p(x_t|x_0) = N(x_t; (1-t)x_0 + t\mu_1, t^2\sigma_1^2 I)$. Since the perturbation kernel is Gaussian, following Song et al. (2020), we assume that the drift term is affine; i.e. $f(x, t) \equiv f(t)x$. Further since $X_0, X_1$ are independent, we can directly infer the first and second moments $\mu_t, \Sigma_t$ for the marginals $p_t(x)$ as $\mu_t = (1-t)\mu_0 + t\mu_1$ and $\Sigma_t = (1-t)^2\Sigma_0 + t^2\sigma_1^2 I$.

From Eq (5.50) of Särkkä & Solin (2019) we have

$$\frac{d\mu_t}{dt} = \mathbb{E}_{p_t(x)}[f(t)x] \tag{16}$$

$$= f(t)\mu_t \tag{17}$$

where $\mu_t$ is the mean at time $t$. Rearranging and integrating both sides:

$$\ln\frac{\mu_t}{\mu_0} = \int_0^t f(s)ds \tag{18}$$

$$\ln\frac{(1-t)\mu_0 + t\mu_1}{\mu_0} = \int_0^t f(s)ds \qquad \text{Substituting } \mu_t = (1-t)\mu_0 + t\mu_1 \tag{19}$$

$$\frac{\mu_1 - \mu_0}{(1-t)\mu_0 + t\mu_1} = f(t) \qquad \text{Differentiating both sides w.r.t. } t \tag{20}$$

$$\tag{21}$$

Substituting $\mu_1 = 0$, we get as in Equation (7):

$$f(x, t) = -\frac{x}{1-t} \tag{22}$$

Similarly, from Eq. (5.51) of Särkkä & Solin (2019):

$$\frac{d\Sigma_t}{dt} = \mathbb{E}_{p_t(x)}\left[f(x, t)(x - \mu_t)^T + (x - \mu_t)f(x, t)^T + G(x, t)QG(x, t)^T\right] \tag{23}$$

Substituting $Q \equiv I$ (we are assuming isotropic dispersion), $G(x, t) \equiv g(t)I$ (symmetric, time-dependent diffusion coefficient), and $f(x, t)$ from Equation (22):

$$\frac{d\Sigma_t}{dt} = \mathbb{E}_{p_t(x)}\left[-\frac{x}{1-t}(x - \mu_t)^T - (x - \mu_t)\frac{x^T}{1-t} + g^2(t)I\right] \tag{24}$$

$$= \frac{2}{1-t}\mathbb{E}_{p_t(x)}\left[-xx^T + \mu_t\mu_t^T\right] + g^2(t)I \tag{25}$$

$$= -\frac{2\Sigma_t}{1-t} + g^2(t)I \tag{26}$$

$$\implies \frac{d\Sigma_t}{dt} + \frac{2\Sigma_t}{1-t} = g^2(t)I \tag{27}$$

Above is an inhomogenous differential equation. The integrating factor $I(t)$ can be calculated as:

$$I(t) = \exp\left(\int_0^t \frac{2}{1-s}ds\right) = \frac{1}{(1-t)^2} \tag{28}$$

Multiplying both sides of Equation (27), we can write:

$$\frac{d}{dt}\left[\frac{\Sigma_t}{(1-t)^2}\right] = \frac{g^2(t)I}{(1-t)^2} \tag{29}$$

Integrating both sides:

$$\left[\frac{\Sigma_s}{(1-s)^2}\right]_0^t = \int_0^t \frac{g^2(s)I}{(1-s)^2}ds \tag{30}$$

$$\frac{\Sigma_t}{(1-t)^2} - \Sigma_0 = \int_0^t \frac{g^2(s)I}{(1-s)^2}ds \tag{31}$$

Substituting $\Sigma_t = (1-t)^2\Sigma_0 + t^2\sigma_1^2 I$:

$$\frac{(1-t)^2\Sigma_0 + t^2\sigma_1^2 I}{(1-t)^2} - \Sigma_0^2 = \int_0^t \frac{g^2(s)I}{(1-s)^2}ds \tag{32}$$

Differentiating both sides w.r.t. $t$ and simplifying yields:

$$g^2(t) = \frac{2t\sigma_1^2}{1-t} \tag{33}$$

Substituting $\sigma_1 = 1$ to the result in Equation (7):

$$g(t) = \sqrt{\frac{2t}{1-t}} \tag{34}$$

## B  SCORE FUNCTION FROM RECTIFIED FLOW

Given a base data distribution $p(x)$ and a conditional noising distribution $p_\sigma(\tilde{x}|x)$, Denoising score matching Vincent (2011) learns the score for the marginal $p_\sigma(\tilde{x})$ by optimizing:

$$\nabla_{\tilde{x}}\ln p_\sigma(\tilde{x}) = \arg\min_\psi \mathbb{E}_{p_\sigma(x_0,\tilde{x})}\left[\frac{1}{2}\left\|\psi(\tilde{x}) - \frac{\partial\ln p_\sigma(\tilde{x}|x_0)}{\partial\tilde{x}}\right\|^2\right] \tag{35}$$

where $p_\sigma(x_0,\tilde{x}) \equiv p(x)p_\sigma(\tilde{x}|x_0)$. The solution to the above optimization problem can be written as:

$$\nabla_{\tilde{x}}\ln p_\sigma(\tilde{x}) = \mathbb{E}_{p_\sigma(x_0|\tilde{x})}\frac{\partial\ln p_\sigma(\tilde{x}|x_0)}{\partial\tilde{x}} \tag{36}$$

Mapping the above to rectified flow with $\sigma \equiv t, \tilde{x} \equiv x_t$ we get:

$$\nabla_{x_t}\ln p_t(x_t) = \mathbb{E}_{p_t(x_0|x_t)}\frac{\partial\ln p_t(x_t|x_0)}{\partial x_t} \tag{37}$$

Next if:

$$p_t(x_t|x_0) = N(x_t; \mu(x_0,t), \sigma^2(x_0,t)I) \tag{38}$$

$$\frac{\partial\ln p_t(x_t|x_0)}{\partial x_t} = \frac{\partial}{\partial x_t}\frac{-\|x_t - \mu(x_0,t)\|^2}{2\sigma(x_0,t)^2} \tag{39}$$

$$= \frac{-(x_t - \mu(x_0,t))}{\sigma(x_0,t)^2} \tag{40}$$

Now:

$$\nabla_{x_t}\ln p_t(x_t) = \mathbb{E}_{p_t(x_0|x_t)}\frac{-(x_t - \mu(x_0,t))}{\sigma(x_0,t)^2} \tag{41}$$

Next, assume the covariance $\sigma(x_0,t)$ doesn't depend on $x_0$ – i.e., $\sigma(x_0,t) \equiv \sigma(t)$ – and the mean $\mu(x_0,t)$ is linear in $x_0$. Then:

$$\mathbb{E}_{p_t(x_0|x_t)}\frac{-(x_t - \mu(x_0,t))}{\sigma(x_0,t)^2} = \mathbb{E}_{p_t(x_0|x_t)}\frac{-(x_t - \mu(x_0,t))}{\sigma(t)^2} \tag{42}$$

$$= \frac{-(x_t - \mu(\mathbb{E}[x_0|x_t],t))}{\sigma(t)^2} \tag{43}$$

## B.1 GAUSSIAN RECTIFIED FLOW

Consider the special case where $x_t = (1-t)x_0 + tx_1, x_1 \sim N(x_1; \mu_1, \sigma_1^2 I)$. We have $p_t(x_t|x_0) = N((1-t)x_0 + t\mu_1, t^2\sigma_1^2)$. Using the result from Equation (43) we get:

$$\nabla_{x_t} \ln p_t(x_t) = \frac{-(x_t - \mu(\mathbb{E}[x_0|x_t], t))}{\sigma(t)^2} \tag{44}$$

From this, we can write:

$$x_1 = \frac{x_t - (1-t)x_0}{t} \tag{45}$$

$$\mathbb{E}[x_1 - x_0|x_t] = \mathbb{E}\left[\frac{x_t - (1-t)x_0}{t} - x_0 \Big| x_t\right] \tag{46}$$

$$= \mathbb{E}\left[\frac{x_t - (1-t)x_0 - tx_0}{t} \Big| x_t\right] \tag{47}$$

$$= \mathbb{E}\left[\frac{x_t - x_0}{t} \Big| x_t\right] \tag{48}$$

$$= \frac{x_t - \mathbb{E}[x_0|x_t]}{t} \tag{49}$$

$$\mathbb{E}[x_0|x_t] = x_t - t\mathbb{E}[x_1 - x_0|x_t] \tag{50}$$

$$\nabla_{x_t} \ln p_t(x_t) = \frac{\mu(\mathbb{E}[x_0|x_t], t) - x_t}{\sigma(t)^2} \tag{51}$$

$$= \frac{(1-t)\mathbb{E}[x_0|x_t] + t\mu_1 - x_t}{t^2\sigma_1^2} \tag{52}$$

$$= \frac{(1-t)(x_t - t\mathbb{E}[x_1 - x_0|x_t]) + t\mu_1 - x_t}{t^2\sigma_1^2} \tag{53}$$

$$= \frac{-(1-t)t\mathbb{E}[x_1 - x_0|x_t] + t\mu_1 - tx_t}{t^2\sigma_1^2} \tag{54}$$

$$= \frac{-(1-t)\mathbb{E}[x_1 - x_0|x_t] + \mu_1 - x_t}{t\sigma_1^2} \tag{55}$$

## B.2 GENERAL RECTIFIED FLOW

First recall the change of variables formula for a density $p(x)$ with $y = g(x)$ where $g$ is invertible and $g^{-1}$ is differentiable:

$$p(y) = p(g^{-1}(y)) \left| \det\left[\frac{\partial g^{-1}(z)}{\partial z}\right]_{z=y}\right| \tag{56}$$

Now, with $x_0 \sim p_1(x_0)$ and $x_1 \sim p_1(x_1)$ and $x_0, x_1 \in \mathbb{R}^d$, let $x_t = g(x_1; x_0)$ be a function that is invertible in first argument and whose inverse $g^{-1}(x_t; x_0)$ is differentiable w.r.t. the first argument. Note that for simple rectified flows, $x_t = (1-t)x_0 + tx_1$ satisfies these conditions.

We can now express the conditional density $p(x_t|x_0)$ as:

$$p(x_t|x_0) = p_1(g^{-1}(x_t; x_0)) \left|\det\left[\nabla_z g^{-1}(z; x_0)\right]_{z=x_t}\right| \tag{57}$$

The score for the conditional density can then be calculated as

$$\frac{\partial \ln p_t(x_t|x_0)}{\partial x_t} = \nabla_z \ln p_1(z)|_{z=g^{-1}(x_t; x_0)} + \nabla_{x_t} \ln \left|\det\left[\nabla_z g^{-1}(z; x_0)\right]_{z=x_t}\right| \tag{58}$$

and the score for the marginal density as:

$$\nabla_{x_t} \ln p_t(x_t) = \mathbb{E}_{p_t(x_0|x_t)} \frac{\partial \ln p_t(x_t|x_0)}{\partial x_t} \tag{59}$$

$$= \mathbb{E}_{p_t(x_0|x_t)} \left[\nabla_z \ln p_1(z)|_{z=g^{-1}(x_t; x_0)} + \nabla_{x_t} \ln \left|\det\left[\nabla_z g^{-1}(z; x_0)\right]_{z=x_t}\right|\right] \tag{60}$$

For the specific case of Rectified flows, define $g(x_1; x) = (1-t)x + tx_1$. Then:

$$x_t = g(x_1; x_0) \tag{61}$$

$$g^{-1}(x_t; x_0) = \frac{x_t - (1-t)x_0}{t} \qquad\qquad \text{Inverse is w.r.t. first argument} \tag{62}$$

$$\frac{\partial g^{-1}(x_t; x_0)}{\partial x_t} = \frac{1}{t}I \tag{63}$$

$$\det \frac{1}{t}I = \frac{1}{t^d} \qquad\qquad I \text{ is } d \times d \tag{64}$$

$$p_t(x_t|x_0) = \frac{1}{t^d}p_1\left(\frac{x_t - (1-t)x_0}{t}\right) \tag{65}$$

Substituting into Equation (60):

$$\nabla_{x_t} \ln p_t(x_t) = \mathbb{E}_{p_t(x_0|x_t)}\left[\frac{1}{t}\nabla_z \ln p_1(z)|_{z=g^{-1}(x_t;x_0)}\right] \tag{66}$$

It can be verified that with the choice of $p_1(x_1) \equiv N(\mu_1, \sigma_1^2 I)$, we recover Equation (55).

## C  PROOF OF THEOREM 1

**Theorem 1.** *Let $p_t(x)$ be the probability density of the solutions of the SDE in Equation* (4) *evolving as $\frac{\partial p_t}{\partial t}$. Then, the probability density of solutions of the following set of SDEs, indexed by $\tilde{G}, \gamma_t$, also evolves as $\frac{\partial p_t}{\partial t}$.*

$$dx = \bar{f}(x,t)dt + \bar{G}(x,t)dW_t \tag{8}$$

*where*

$$\bar{f} = f - \frac{1}{2}\left(\nabla \cdot [(1-\gamma_t)GG^T - \tilde{G}\tilde{G}^T] + [(1-\gamma_t)GG^T - \tilde{G}\tilde{G}^T] \cdot \nabla \ln p_t\right) \tag{9}$$

$$\bar{G} = [\gamma_t GG^T + \tilde{G}\tilde{G}^T]^{\frac{1}{2}} \tag{10}$$

*and $\tilde{G} \equiv \tilde{G}(x,t), \gamma_t \geq 0$ are arbitrary functions such that the solutions of Equation* (8) *exist and are unique.*

*Proof.* The evolution of the marginal probability density $p_t(x)$ is then described by the Fokker-Planck-Kolmogorov (FPK) equation (Särkkä & Solin, 2019) as:

$$\frac{\partial p_t}{\partial t} = -\sum_{i=1}^{d}\frac{\partial}{\partial x_i}[\bar{f}p_t] + \frac{1}{2}\sum_{i=1}^{d}\sum_{j=1}^{d}\frac{\partial^2}{\partial x_i x_j}\left[\sum_{k=1}^{d}\bar{G}_{ik}\bar{G}_{jk}p_t\right] \tag{67}$$

We write the above more succinctly as:

$$\frac{\partial p_t}{\partial t} = -\nabla \cdot [\bar{f}p_t] + \frac{1}{2}\nabla \cdot [\bar{G}\bar{G}^T p_t] \cdot \nabla^T \tag{68}$$

Where $\nabla\cdot$ is the divergence operator. Next for an arbitrary $R \equiv R(x,t)$ consider the following identity:

$$[RR^T p_t] \cdot \nabla^T = \nabla \cdot [RR^T p_t] \qquad\qquad RR^T \text{ is symmetric} \tag{69}$$

$$= [\nabla \cdot RR^T]p_t + RR^T \cdot \nabla p_t \tag{70}$$

$$= [\nabla \cdot RR^T]p_t + RR^T \cdot p_t\nabla \ln p_t \tag{71}$$

$$= (\nabla \cdot RR^T + RR^T \cdot \nabla \ln p_t)p_t \tag{72}$$

Expanding out $\bar{f}p_t$ by substituting for $\bar{f}$:

$$\bar{f}p_t = \left[f - \frac{1}{2}\left(\nabla \cdot [(1-\gamma_t)GG^T - \tilde{G}\tilde{G}^T] + [(1-\gamma_t)GG^T - \tilde{G}\tilde{G}^T] \cdot \nabla \ln p_t\right)\right]p_t \quad (73)$$

$$= fp_t - \frac{1-\gamma_t}{2}\left(\nabla \cdot GG^T + GG^T \cdot \nabla \ln p_t\right)p_t$$
$$+ \frac{1}{2}\left(\nabla \cdot \tilde{G}\tilde{G}^T + \tilde{G}\tilde{G}^T \cdot \nabla \ln p_t\right)p_t \quad (74)$$

Using Equation (72) and rewriting:

$$\bar{f}p_t = fp_t - \frac{1-\gamma_t}{2}[GG^T p_t] \cdot \nabla^T + \frac{1}{2}[\tilde{G}\tilde{G}^T p_t] \cdot \nabla^T \quad (75)$$

Next we revisit Equation (68), and substitute for $\bar{f}p_t$ and $\bar{G}$ with $\bar{G}\bar{G}^T = \gamma_t GG^T + \tilde{G}\tilde{G}^T$:

$$\frac{\partial p_t}{\partial t} = -\nabla \cdot [fp_t - \frac{1-\gamma_t}{2}[GG^T p_t] \cdot \nabla^T + \frac{1}{2}[\tilde{G}\tilde{G}^T p_t] \cdot \nabla^T]$$
$$+ \frac{1}{2}\nabla \cdot \left[(\gamma_t GG^T + \tilde{G}\tilde{G}^T)p_t\right] \cdot \nabla^T \quad (76)$$

$$= -\nabla \cdot [fp_t] + \frac{1-\gamma_t}{2}\nabla \cdot [GG^T p_t] \cdot \nabla^T - \frac{1}{2}\nabla \cdot [\tilde{G}\tilde{G}^T p_t] \cdot \nabla^T$$
$$+ \frac{\gamma_t}{2}\nabla \cdot [GG^T p_t] \cdot \nabla^T + \frac{1}{2}\nabla \cdot \left[\tilde{G}\tilde{G}^T p_t\right] \cdot \nabla^T \quad (77)$$

With cancellations, we arrive at:

$$\frac{\partial p_t}{\partial t} = -\nabla \cdot [fp_t] + \frac{1}{2}\nabla \cdot [GG^T p_t] \cdot \nabla^T \quad (78)$$

which is the FPK equation describing the time evolution of the marginal probability density $p_t(x)$ of the solutions of the SDE in Equation (4).

$\square$

## D    PROOF OF COROLLARY 1.1

**Corollary 1.1.** *For the SDE in Equation* (4) *with* $G \equiv g(t)I$, *a subset of SDEs prescribed by Theorem 1 and indexed by* $\gamma_t$ *is:*

$$dx = \left[f(x,t) - \frac{(1-\gamma(t))g^2(t)}{2}\nabla_x \ln p_t(x)\right]dt + \sqrt{\gamma(t)}g(t)dW_t \quad (11)$$

*Proof.* Starting with Theorem 1, let's define $\tilde{G} \equiv 0$ and $G \equiv g(t)I$, where $g(t)$ is a scalar valued function. These choices lead to following

$$\bar{G} = [\gamma_t(g(t)I)^2]^{\frac{1}{2}} = \sqrt{\gamma_t}g(t)I \quad (79)$$

$$\bar{f} = f - \frac{1}{2}\left(\nabla \cdot [(1-\gamma_t)g^2(t)I] + [(1-\gamma_t)g^2(t)I] \cdot \nabla \ln p_t\right) \quad (80)$$

$$= f - \frac{1}{2}\left([(1-\gamma_t)g^2(t)I] \cdot \nabla \ln p_t\right) \quad (81)$$

$$= f - \frac{(1-\gamma_t)g^2(t)}{2}\nabla \ln p_t \quad (82)$$

Note that Equation (81) follows from $\nabla \cdot [(1-\gamma_t)g^2(t)I] = 0$ since neither $\gamma_t$ nor $g(t)$ are functions of $x$. $\square$

# E  PROOF OF COROLLARY 1.2

**Corollary 1.2.** *For the ODE in Equation* (3)*, a subset of SDEs prescribed by Theorem 1 and indexed by $\tilde{g}(t)$ is*

$$dx = \left[ v(x,t) + \frac{\tilde{g}^2(t)}{2} \nabla_x \ln p_t(x) \right] dt + \tilde{g}(t) dW_t \tag{12}$$

*Proof.* First note that $f \equiv v(x,t)$ by definition from equation (3) in the paper. Now, again starting with Theorem 1, let's define $\tilde{G} \equiv \tilde{g}(t)I$ and $G \equiv 0$, where $\tilde{g}(t)$ is a scalar valued function. These choices lead to following

$$\bar{G} = [(\tilde{g}(t)I)^2]^{\frac{1}{2}} = \tilde{g}(t)I \tag{83}$$

$$\bar{f} = v(x,t) - \frac{1}{2} \left( \nabla \cdot [-\tilde{g}^2(t)I] + [-\tilde{g}^2(t)I] \cdot \nabla \ln p_t \right) \tag{84}$$

$$= v(x,t) + \frac{1}{2} \left( [\tilde{g}^2(t)I] \cdot \nabla \ln p_t \right) \tag{85}$$

$$= v(x,t) + \frac{\tilde{g}^2(t)}{2} \nabla \ln p_t \tag{86}$$

Note that Equation (85) follows from $\nabla \cdot [-\tilde{g}^2(t)I] = 0$ since $\tilde{g}(t)$ is not a function of $x$. $\quad\square$

# F  CLOSED FORM RECTIFIED FLOW EXPRESSION FOR THE TWO GAUSSIAN CASE

Our empirical studies use a two Gaussian toy problem setup. We state the closed form expression for the rectified flow for this case. Consider $x_0 \sim N(\mu_0, \sigma_0^2 I), x_1 \sim N(\mu_1, \sigma_1^2 I)$:

$$x_t = \alpha_t x_0 + \beta_t x_1, \quad \alpha_t > 0, \alpha_0 = 1, \alpha_1 = 0, \beta_t > 0, \beta_0 = 0, \beta_1 = 1 \tag{87}$$

The marginal density $p_t(x_t)$ is also Gaussian:

$$p_t(x_t) = N(x_t; \alpha_t \mu_0 + \beta_t \mu_1, \alpha_t^2 \sigma_0^2 + \beta_t^2 \sigma_1^2) \tag{88}$$

We have:

$$v(x,t) = \mathbb{E}[x_1 - x_0 | x_t] \equiv \mathbb{E}_{p(x_0, x_1 | x_t)}[x_1 - x_0] \tag{89}$$

Using the following:

$$p(x_0, x_1 | x_t) = \frac{p(x_t | x_0, x_1) p(x_0, x_1)}{p(x_t)} = \frac{p(x_t | x_0, x_1) p_0(x_0) p_1(x_1)}{p(x_t)} \tag{90}$$

$$p(x_t | x_0, x_1) = \delta(x_t - (1-t)x_0 - t x_1) \tag{91}$$

and elementary properties of Gaussian and Dirac delta distributions, it can be verified that:

$$v(x,t) = \frac{(k_t \mu_1 - x_t)\alpha_t \sigma_0^2 + (x_t - k_t \mu_0)\beta_t \sigma_1^2}{\alpha_t^2 \sigma_0^2 + \beta_t^2 \sigma_1^2} \tag{92}$$

where $k_t = \alpha_t + \beta_t$.

# G  TOY GAUSSIAN EXPERIMENT DETAILS

In the experiments in Section 4.1 we study how various SDEs in Table 1 behave on a toy problem where both $p_0 \equiv N(-1, 0.3)$ and $p_1 \equiv N(0, 1.0)$ are Gaussian. In this case the marginal distributions $p_t$ for Gaussian flow are Gaussian with $p_t = N(\mu_t, \sigma_t^2)$ and the true statistics $\mu_t, \sigma_t^2$ can be easily computed. In addition, the rectified flow $v(x,t)$ is available in closed form (see Appendix F). The SDEs are simulated backwards in time from $t = 1$ with draws from $p_1$ using Equation (5). The drift $\tilde{f}$

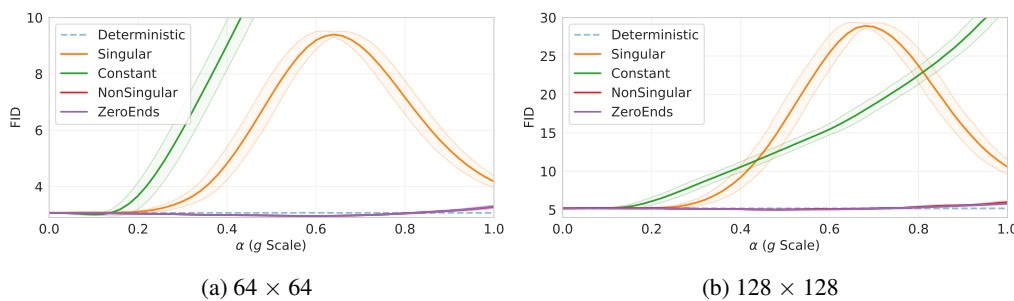

(a) $64 \times 64$            (b) $128 \times 128$

Figure 7: **Non-singular samplers work well over a broad range of** $\alpha$**.** The same plots as Figure 5, but showing a larger range of FIDs. Note that the Singular sampler is highly non-monotonic as a function of $\alpha$.

and diffusion $\tilde{g}$ terms are calculated using Table 1 by setting $\alpha = 1$ and using the closed form $v(x, t)$ from Equation (92). We simulate 10 trials of 10000 trajectories using Euler-Maruyama discretization with varying number of steps. Estimates for mean $\mu_t$ and variance $\sigma_t^2$ at each timestep for various SDEs are calculated, along with their standard deviation across trials. Error, calculated as estimate - truth, is then plotted in Figures 2 to 4 for both the mean and the variance estimates along with the KL-Divergence from the true marginal distribution.

## H  IMAGENET EXPERIMENT TRAINING/EVALUATION DETAILS

We train two base Rectified flow models to yield $v(x, t)$ at two resolutions of $64 \times 64$ and $128 \times 128$, on the entire ImageNet training dataset containing roughly 1.2 million images. Our model is based on the architecture described in Hoogeboom et al. (2023). The model is structured such that the lower feature map resolution is $16 \times 16$. Therefore, for $64 \times 64$ resolution two downsamplings are performed, while for $128 \times 128$ three downsamplings are performed. The model is trained with SGD using adamw (Kingma & Ba, 2014; Loshchilov & Hutter, 2017) with $\beta_1 = 0.9, \beta_2 = 0.99, \epsilon = 10^{-12}$ for 1000 epochs. We use center crop and left-right flips as the only augmentations. An exponential moving average, with a decay of 0.9999, of parameters is used for evaluation. FIDs are reported over the training dataset with reference statistics computed with center crop but without any augmentation, but with class conditioning. Samplers were evaluated for all $\alpha \in \{0.0, 0.02, 0.04, \ldots, 1.0\}$.

The $64 \times 64$ model trained for 500 epochs in 4 days, 8 hours on $8 \times 8$ Google Cloud TPUs v3. The $128 \times 128$ model trained for 500 epochs in 4 days, 20 hours on $8 \times 8$ Google Cloud TPUs v3.

## I  EXAMPLE IMPLEMENTATION

See Figure 8 for an example implementation of the NonSingular sampler.

## J  ADDITIONAL EXPERIMENTAL RESULTS

### J.1  FID VS $\alpha$

In Figure 7 we show a larger range FID for various samplers compared in Figure 5. We observe that the Singular sampler tends to perform well only at low scales with an intriguing behavior for higher scales where the FID starts to improve again after worsening significantly.

### J.2  EFFECT OF DIFFUSION COEFFICIENT MAGNITUDE ON SAMPLES

We qualitatively visualize the effect of diffusion coefficient magnitude for the three SDEs discussed in the main paper. Figure 9 visualizes samples for the constant diffusion term SDE as a function increasing coefficient magnitude. Each column is a different sample starting with the same random

Figure 8: NonSingular Sampler written in JAX.

```
1  def non_singular_sampler(
2      rng, num_samples, model, params, labels, g_scale, num_steps=1000,
3      batch_size=10, image_size=64, num_channels=3, num_classes=1000,
4      n=1, m=0):
5    """Draw samples from the model."""
6    p_1_samples = []
7    p_0_samples = []
8    t = jnp.linspace(1., 0., num_steps+1)
9    t_ones = jnp.ones([batch_size, 1, 1])
10
11   # Sampler loop body
12   def body_fn(i, z):
13     z, labels, rng = z
14     tb = t[i] * t_ones
15     z_hat = model.apply({'params': params}, z, (1 - tb), labels)
16     v = -z_hat
17     b = g_scale
18     g = b * jnp.power(tb, n / 2) * jnp.power(1 - tb, m / 2)
19     s_u = -((1-tb) * v + z)
20     fr = (v - jnp.square(b) * jnp.power(tb, n-1
21             * jnp.power(1-tb, m) * s_u / 2)
22     rng, key = jax.random.split(rng)
23     dt = t[i+1] - t[i]
24     dbt = (jnp.sqrt(jnp.abs(dt))
25             * jax.random.normal(key, shape=z.shape))
26     z = z + fr * dt + g * dbt
27     return z, labels, rng
28
29   max_steps = num_samples // batch_size
30   for _ in range(max_steps):
31     # Sample from p_1
32     rng, key = jax.random.split(rng)
33     z = sample_from_prior(
34       key, shape=[batch_size, image_size, image_size, num_channels])
35     p_1_samples.append(z)
36
37     # Run the sampler
38     rng, key = jax.random.split(rng)
39     init_val = (z, labels, key)
40     z, _, _ = jax.lax.fori_loop(
41         lower=0, upper=num_steps, body_fun=body_fn, init_val=init_val)
42     p_0_samples.append(z)
43
44   p_1_samples = jnp.concatenate(p_1_samples, axis=0)
45   p_0_samples = jnp.concatenate(p_0_samples, axis=0)
46   return p_1_samples, p_0_samples
```

draw from $p_1(x_1)$. Each row corresponds to a different magnitude for the diffusion coefficient $g(t)$. Figures 10 and 11 visualize samples with a similar scheme for the non-singular and singular SDE.

### J.3  CLASSIFIER-FREE GUIDANCE SAMPLES

Figure 12 shows additional samples using classifier-free guidance with NonSingular at different values of $\alpha$ and $\lambda$, as in Figure 1.

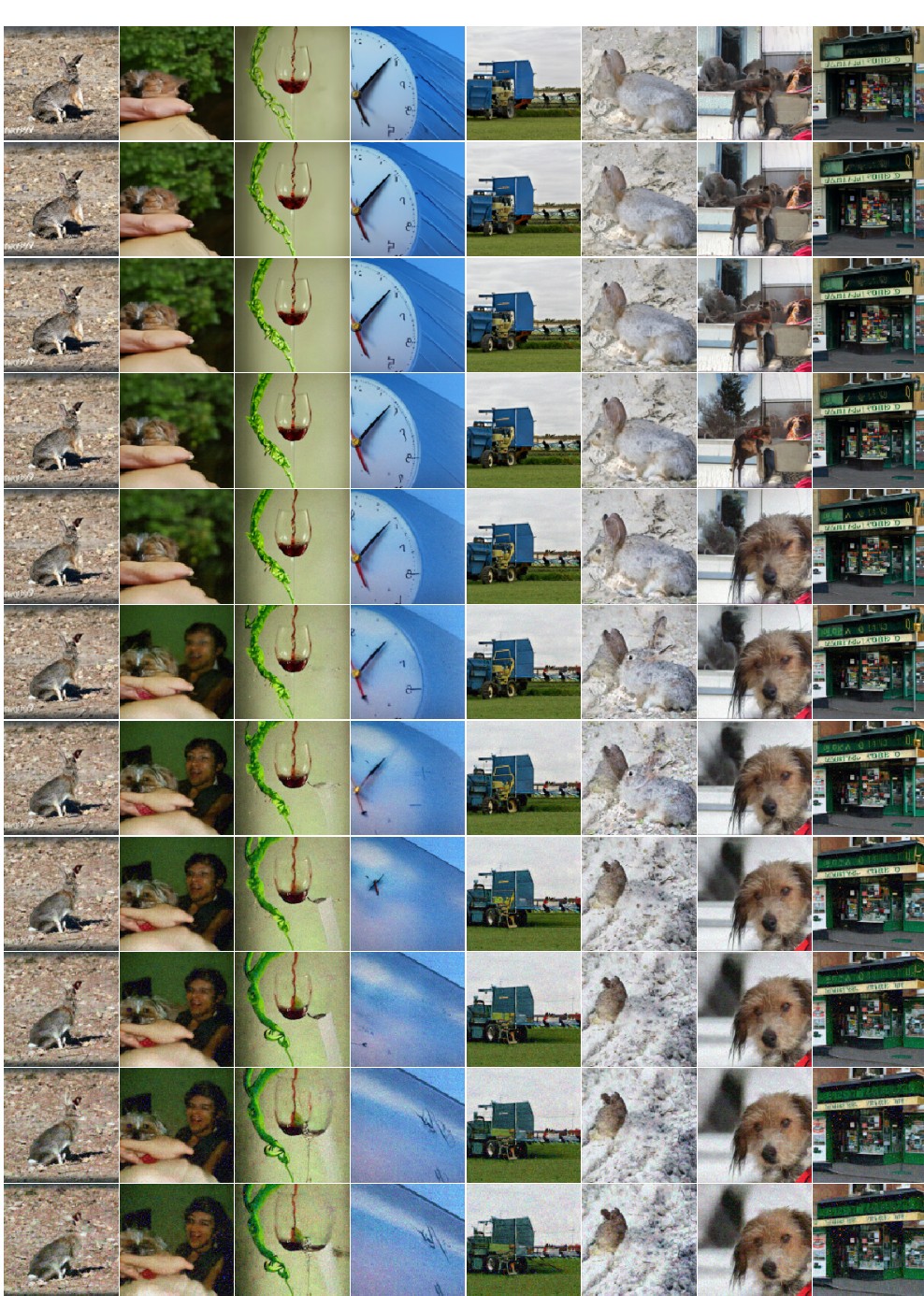

Figure 9: **Constant samples with increasing scaling** $\alpha$. Each row displays samples at a particular $g$-scale, from $0$ increasing to $1$ in the increments of $0.1$ from top to bottom. Sampling for each columns starts off with the same initial noise image and conditioning class.

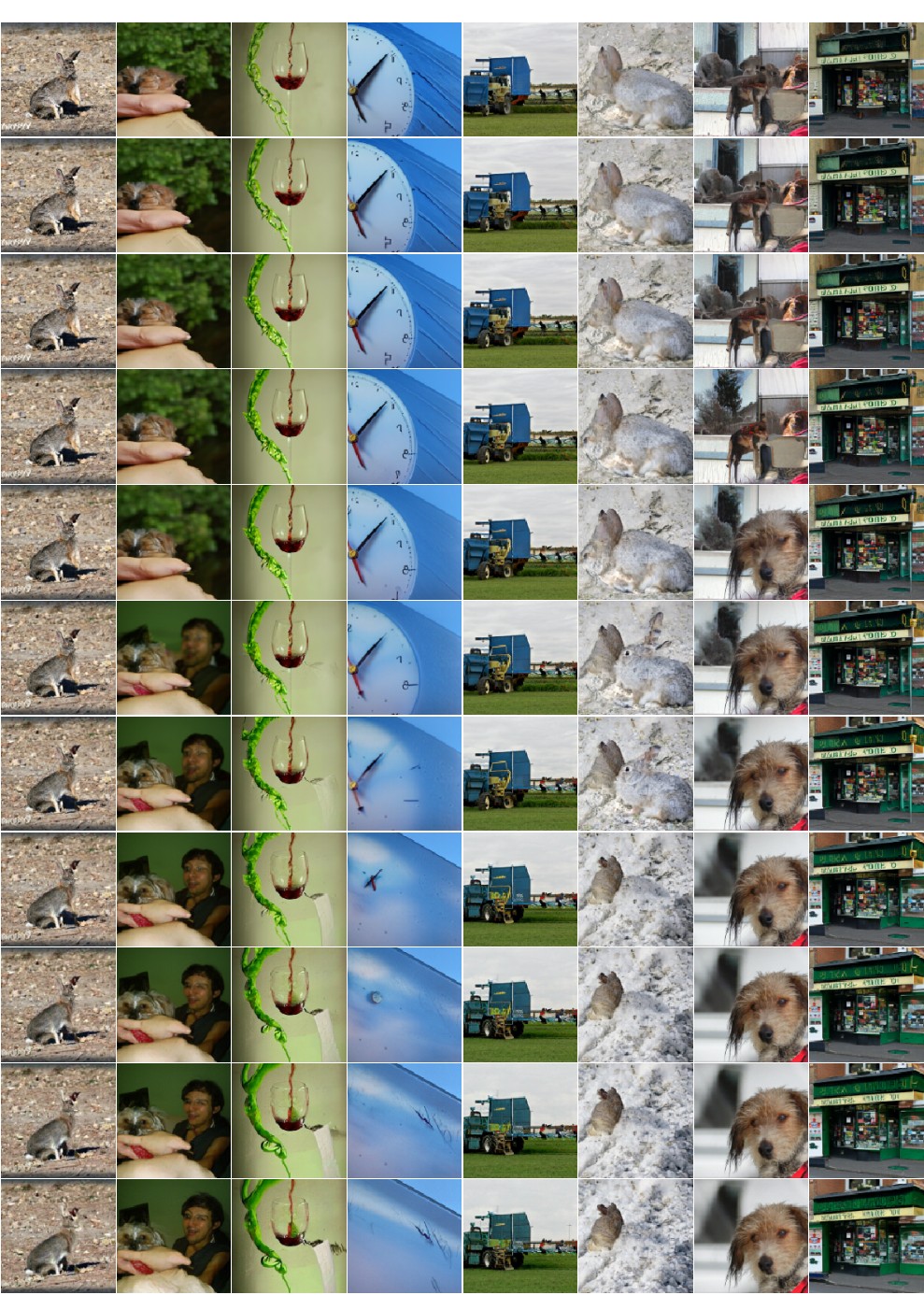

Figure 10: **NonSingular samples with increasing scaling** $\alpha$. Each row displays samples at a particular $g$-scale, from $0$ increasing to $1$ in the increments of $0.1$ from top to bottom. Sampling for each columns starts off with the same initial noise image and conditioning class.

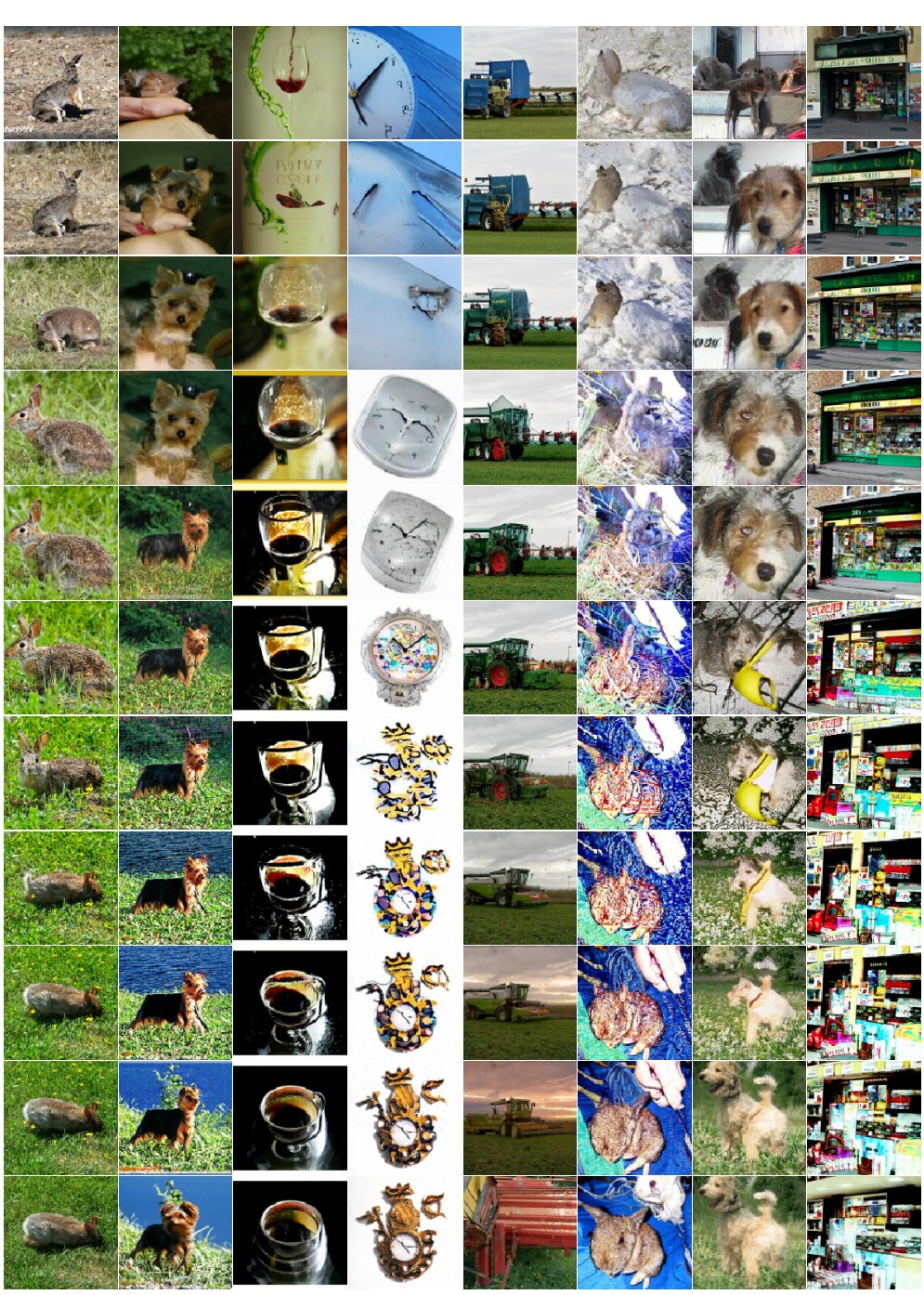

Figure 11: **Singular samples with increasing scaling** $\alpha$**.** Each row displays samples at a particular $g$-scale, from $0$ increasing to $1$ in the increments of $0.1$ from top to bottom. Sampling for each columns starts off with the same initial noise image and conditioning class.

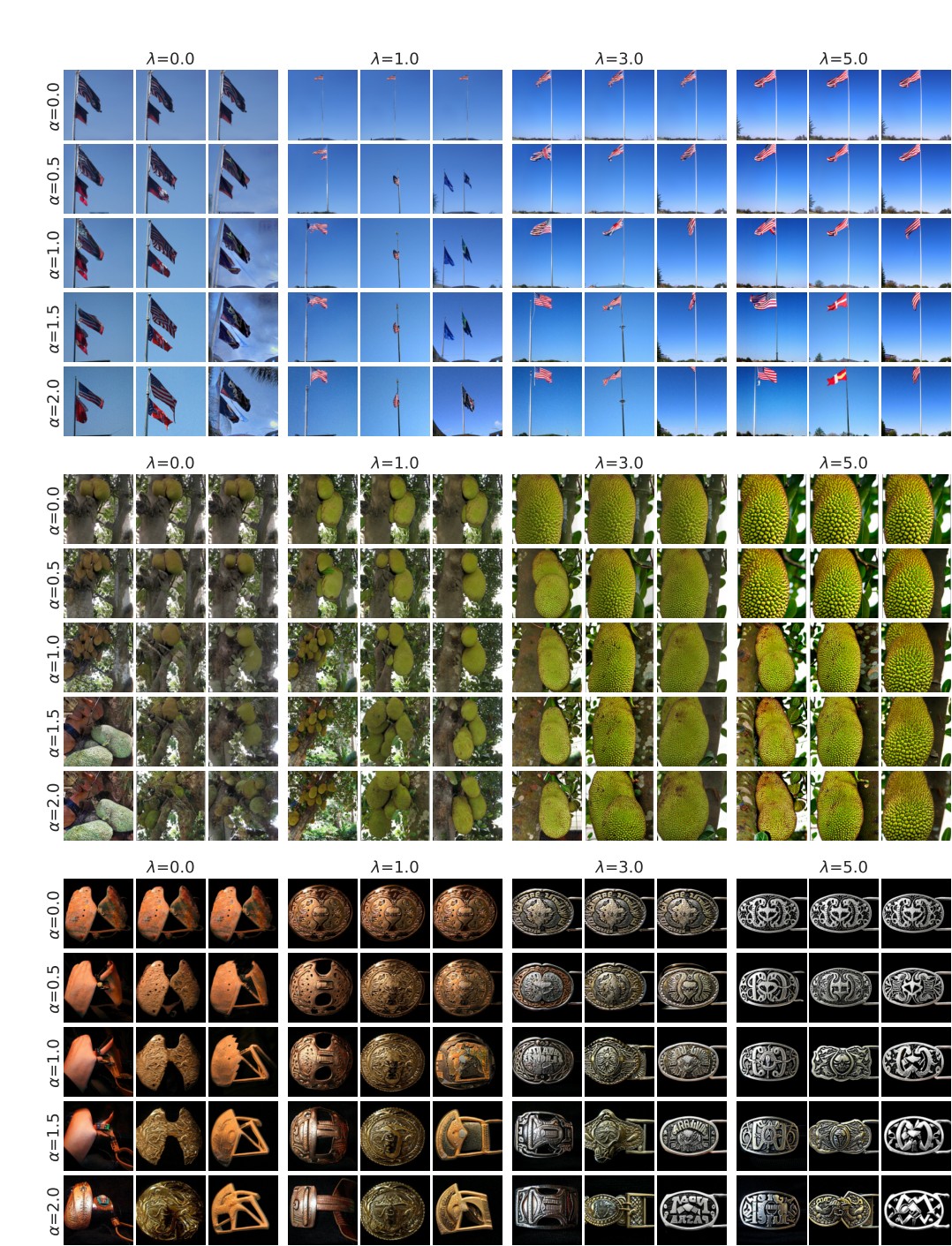

Figure 12: **Stochastic sampling improves diversity at all classifier-free guidance levels.** Additional results as in Figure 1, described in Section 4.2.

