# OpenReview forum: "Stochastic Sampling from Deterministic Flow Models"
_ICLR.cc/2025/Conference — Submitted to ICLR 2025_

### Official Review · Reviewer_jr5u · 2024-10-27

**Soundness:** 4
**Presentation:** 4
**Contribution:** 3
**Rating:** 8
**Confidence:** 4

**Summary:**

This paper introduces a way to take a trained deterministic flow model and find a family of stochastic flow models that induce the same marginals and which can be simulated using the original learned deterministic flow. This allows one to sample from a much broader range of flow models targeting the same distribution without retraining the network.

The authors claim and then demonstrate that this can be desirable, since stochastic flows have been demonstrated to have better empirical performance in many settings. In particular, the authors show that their stochastic flows have perform better at estimating the variance of the target distribution in a toy setting, and have better FID score when targeting the ImageNet distribution.

**Strengths:**

- The paper's central observation, given in Theorem 1, that a learned deterministic flow can be converted into a family of stochastic flows targeting the same distribution is a neat theoretical result. It also has obvious practical relevance since, as the authors claim, stochastic sampling methods tend to perform better in practice.
- The presentation of the paper is extremely clear throughout, and I had no trouble understanding the authors arguments and contributions. Thank you for making it easy to engage with your work!
- The authors provide a compelling empirical demonstration of the practical success of their method, on both toy and real-world data distributions.

Overall, I believe this to be a very technically solid paper worthy of acceptance to ICLR. While the core idea of the contribution is not totally novel - the idea that one can convert between SDE and ODE sampling methods via rearranging the Fokker-Plank PDEs governing the marginals is implicit in previous work - this is the first work I'm aware of that demonstrates this practically in the ODE to SDE direction. This is a useful idea and I find the authors' empirical demonstrations of its relevance compelling.

**Weaknesses:**

There are no significant weaknesses with this paper that I would want to raise. I have just a couple of minor editorial points:
- $\bar G$ and $\tilde G$ are very hard to distinguish in print, and you might want to consider alterantive notation here.
- It took me a while to understand the plots in Figures 1, 2 and 3; for example what exactly was meant by "error in mean" and "error in variance", and which distributions you were measuring the KL between. If there's an easy way to clarify this, that might be worth adding.

**Questions:**

None in addition to the points raise above.

---

> ### Author Response · Authors · 2024-11-22
> **Thank you for your feedback**
>
> We highly appreciate your positive assessment and encouragement. We will improve the notation and also the text/captions to make the Figures 1, 2, 3 more understandable.

---

### Official Review · Reviewer_7s8u · 2024-10-28

**Soundness:** 3
**Presentation:** 3
**Contribution:** 1
**Rating:** 3
**Confidence:** 5

**Summary:**

The paper introduces a method to enhance deterministic flow models, such as rectified flows, by converting their underlying ordinary differential equations (ODEs) into a family of stochastic differential equations (SDEs) with the same marginal distributions. This approach allows for a continuous transition between deterministic and stochastic sampling, using both the flow field and the score function. The method addresses limitations of deterministic samplers, improving their performance on tasks like Gaussian modeling and large-scale ImageNet generation. Additionally, it offers a mechanism to control the diversity of generated samples, providing more flexibility in the sampling process.

**Strengths:**

1. This paper is well-written, with thorough ablation studies that carefully compare deterministic and stochastic samplers, which I greatly appreciate.

2. The paper is well-written and easy to follow.

3. The motivation is clear.

**Weaknesses:**

1.To the best of my knowledge, the paper lacks novelty. The core result of Theorem 1 appears straightforward to derive by combining existing results (Equation 4 and Appendix D in [1], Doob's h-transform, and Equation 37 in [2]).

Let me explain more. The Theorem 1 can be understood as

$\bar{f}=f-\frac{1}{2}[\nabla(GG^T-(\gamma_t GG^T+\tilde{G}\tilde{G}^T)]-\frac{1}{2}[GG^T-(\gamma_t GG^T+\tilde{G}\tilde{G}^T)]\nabla \log p$,

$=f-\frac{1}{2}[\nabla(GG^T-\bar{G}\bar{G}^T)]-\frac{1}{2}[GG^T-\bar{G}\bar{G}^T)]\nabla \log p$,
and

$\bar{G}=[\gamma_t GG^T+\tilde{G}\tilde{G}^T]^{1/2}$

According to [1], The original SDE eq.4 can be written as,
$dx_t= [f-\frac{1}{2}\nabla[GG^T]-\frac{1}{2}GG^T\nabla\log p]dt$

Ok then, all the tricks happen in the diffusion term of FPK. You can easily extend eq.28 in [1] with $x_t$ dependent $G(x,t)$ (it may give even more flexibility?), and absorbing the additional term into the drift term as shown in eq.37 in [2].

My understanding is that, once an equivalence is established between flow/bridge models and diffusion models, the remaining work can be readily extended.

However, I want to acknowledge the authors' efforts; despite the challenges of the problem, the community still benefits from having researchers dedicated to addressing it.

[1]Zhang, Qinsheng, and Yongxin Chen. "Fast sampling of diffusion models with exponential integrator." arXiv preprint arXiv:2204.13902 (2022).

[2]Song, Yang, et al. "Score-based generative modeling through stochastic differential equations." arXiv preprint arXiv:2011.13456 (2020).

**Questions:**

I do not have any further questions. And also, feel free to let me know if I misunderstood anything!

---

> ### Author Response · Authors · 2024-11-22
> **Please refer to common response**
>
> Please see the common response above for novelty related concerns.

---

> > ### Comment · Reviewer_7s8u · 2024-11-23
> > **Thanks for the reply**
> >
> > Thanks for the clarification. I have increased confidence.

---

> > > ### Author Response · Authors · 2024-11-27
> > > **Justification of reviewer position**
> > >
> > > While it is natural to anchor and dig-in when faced with a push back, we urge the reviewer to respond with valid arguments for how their position is justified in the light of our responses, how our responses do not address their concerns and what we can further clarify to help address those concerns. Please note that we have posted an additional common response above based on reviewer vUKT’s comments and we urge you to respond there.

---

### Official Review · Reviewer_vUKt · 2024-11-03

**Soundness:** 3
**Presentation:** 3
**Contribution:** 1
**Rating:** 3
**Confidence:** 5

**Summary:**

This paper describes how deterministic dynamical transport models can be changed into stochastic samplers without retraining when the base distribution is Gaussian. They then provide a series of experiments exploring the benefits and trade-offs of using this velocity model under different settings of the SDE: extent of diffusivity and time-dependent schedule of its magnitude.

**Strengths:**

The paper provides a useful empirical exploration of the trade-offs/biases that emerge depending on how the sampling SDE is discretized and which diffusion coefficient is used, e.g. in Table 2.

Section 4.2 itemizes useful takeaways for the tradeoffs in SDE-based sampling, and provides nice experiments to demonstrate how diffusivity alleviates e.g. sample degeneracy.

**Weaknesses:**

First off, thank you for your submission! It would be great if the authors could address the following *primary concern* about the work

Corollary 1 is already well known -- it is one of the main points of the interpolant procedure, and is the essential knob explored in 2303.08797, Corollary 2.18. This was then extensively tested (the variety of potential diffusion coefficients from the same velocity model) in 2401.08740. It's unclear to the reviewer how what this paper is proposing is different from these essential elements. In addition Corollary 1.2 in the given paper is already noticed by Song et al 2011.13456. The authors make remarks about these works in the "related work" section but do not seem to recognize this connection.

Because this is the main point of the paper given in this work, it is unclear to the reviewer what this paper sees as its main contribution to the literature.

**Questions:**

I think it's important for the author's to clarify what the novel contribution is. Could the authors address that?

---

> ### Author Response · Authors · 2024-11-22
> **Please refer to common response**
>
> Please see the common response above for novelty related concerns.

---

### Official Review · Reviewer_Nc6P · 2024-11-04

**Soundness:** 3
**Presentation:** 3
**Contribution:** 1
**Rating:** 3
**Confidence:** 4

**Summary:**

This paper discusses the case of using linear interpolants for training deterministic flow models. The paper motivates using stochastic sampling instead of deterministic sampling.

**Strengths:**

Clearly written.

**Weaknesses:**

Lack of novelty. The conversion between SDEs with different state-independent diffusion coefficients (including zero for ODE) through the Fokker-Planck equation is very well-understood now. While Theorem 1 does cover a general case (state-dependent), this general case is not applied. An example of a paper that discussed stochastic sampling and classifier-free guidance is SiT [1].


[1] https://arxiv.org/abs/2401.08740

**Questions:**

Can the stochastic sampling be generalized to 2-Rectified Flows (i.e. after rectification / training with a non-independent coupling)?

---

> ### Author Response · Authors · 2024-11-22
> **Response to questions**
>
> Please see the common response above for novelty related concerns.
>
> **Q. Can the stochastic sampling be generalized to 2-Rectified Flows …**
>
> Yes! Thank you for this question. It further underlines the value of our work in terms of the different proof techniques employed in our paper, which allow us to answer such non-obvious questions with ease.
>
> For all k-Rectified flows, the same formula for estimating the score function as in eq. 13 and eq. 55 holds, but with the new estimates of the flow field. While not obvious from common proof techniques, this directly follows from our derivation in Appendix B.2 titled "General Rectified Flow". The main assumption needed is that $p_1(x_1)$ continues to be Gaussian so that we can make the necessary substitution in eq. 66 to arrive at eq. 13 and eq. 55. In other words, the assumption is that the reflow procedures don’t lead to a drift in the marginal distribution $p_1(x_1)$.
>
> We will update the paper with a discussion of this additional result. Hopefully this further allays your concerns about the value of our work.

---

### Author Response · Authors · 2024-11-22
**Common response addressing novelty concerns**

While we are thankful to the reviewers for their time reviewing the paper, we are baffled by the reviewer responses which rate “reject” based on a subjective assessment of “lack of novelty”, while simultaneously acknowledging and minimizing the novel aspects of the paper. We would respectfully reiterate that Theorem 1 is our main result (as stated in lines 86-87), and to the best of our knowledge does not occur in literature. It includes various other methods as special cases (including all those mentioned by the reviewers), some of which are previously known (as stated in lines 41, 42, 241, 242 and in additional references provided by the reviewers). Our clear and accessible presentation/writing (as acknowledged by all reviewers) provides an easy to follow recipe. This recipe is more general than those previously reported in the literature, supporting even state-dependent models that do not yet exist in the literature, and permitting transforming in both directions between ODEs and SDEs.

We would request that reviewers apply a concrete rubric when asking whether the work is novel:
- Is this the first occurrence of the contribution in the literature?
- Does it unify and generalize prior work?
- Does it bring new insights and enable new use cases?

If the reviewers reflect on our core contribution and find that the answers to these questions are all yes (we believe they are, and did not see anything in the reviews that contradicts our belief), then please reconsider the rejection ratings for this clearly-presented generalization of a large number of previous works. We commit to citing and discussing reviewer provided new references in the text.

**Reviewer Nc6P:** Theorem 1 is our main theoretical result and includes many existing state independent methods as special cases. We acknowledge that in the paper already (lines 41, 42, 241, 242), and other reviewers point out a few other papers that are a special case that we were unaware of. This only adds value to our main result that presents a unified view of these special cases and provides an easy recipe for a practitioner to make use of. While we do not apply the state dependent case, as all practical models currently use state independent variants, our theoretical result nevertheless holds (proved in appendix) and does not occur in prior art. We do study and ablate state independent variants to verify our core contribution in Theorem 1. Thank you for bringing SiT to our attention. Note that while SiT indeed discusses stochastic sampling and focuses on the special case of one of the end distributions being Gaussian, which we also evaluate in experiments, our Theorem 1 is a more general theoretical result. Further, our proof presented in Appendix B.2 is more general as well, which we exploit to answer your question in the reviewer specific response to you. We will cite SiT and include this discussion in the paper. Would a state dependent toy study further convince the reviewer that Theorem 1 is indeed correct?

**Reviewer vUKt:** As stated in the paper, Theorem 1 is our main theoretical result and includes many existing state independent methods as special cases, and to the best of our knowledge, does not occur in prior art. We derive Corollaries 1.1 and 1.2 as state independent practical special cases, variants of which have been studied in literature (we acknowledge this in paper lines 41, 42, 241, 242). Song et al 2011.13456, do prescribe the SDE -> ODE transformation; in comparison our main result in Theorem 1 allows one to move easily in both directions SDE <-> ODE, while Corollary 1.2 makes the ODE -> SDE case more accessible and obvious (as acknowledged by reviewer jr5u). Further, Theorem 1 makes the inclusion of state dependent cases obvious. We take it as a compliment that our clear writing and presentation makes these cases obvious for the reviewer, and we hope to do the same for a wider audience.

**Reviewer 7s8u:** We are baffled by your assessment of “reject”. In your own state-independent special case proof sketch, you mentioned combining results from at least three different sources ([1]. Doob’s h-transform, [2]), yet you consider our theoretical result to be not novel. One could argue that this is how most research works – we use existing results to build and derive more general novel results. Note that our main theoretical result in Theorem 1 includes the state dependent special case as well. While we do not empirically study it, we are happy to provide empirical evidence that would convince the reviewer of the correctness of Theorem 1. Note that, though our well written and clear presentation makes our results obvious, as evidenced by your assessment, it does not occur in literature. One has to be an expert such as yourself to be able to recall and combine these results when the need arises. We do this for the broader community in our paper and provide an easy to use recipe, accessible to all experts and practitioners alike.

---

> ### Author Response · Authors · 2024-11-27
> **Novelty and Value Statement**
>
> Dear reviewers, thank you for your responses. Following is a statement of the novelty and value addition that our paper does over the existing body of work. We will improve the text to make this more clear.
>
> 1. Theorem 1: This is our main theoretical result and includes many existing state independent methods as special cases, along with generalizing to state dependent cases as well. It presents a simple and unified picture of a variety of existing proposals (state independent), made in different contexts. It makes it easy to derive and explore novel samplers for existing models. We present a simple proof based on FPK equations that is easy to follow without resorting to more advanced stochastic calculus techniques.
> 2. Corollaries 1.1 and 1.2:  We derive and prove two corollaries that directly follow from Theorem 1. One of which (Corollary 1.1) exists in literature (also acknowledged in paper, and pointed out by reviewers). It serves to connect Theorem 1 to existing works, and we arrive at it from a different perspective than those presented in prior art. Corollary 1.2, while superficially similar to eq. 6 in (Song et al. 2011.13456), is actually a bit different. Note the sign difference and a factor of 1/2. Eq. 6 in Song et al. (2011.13456) is arrived at by deriving the time reversal of a given (state independent) SDE. In contrast, Corollary 1.2 is derived as following from Theorem 1, where the starting point is an ODE and therefore it serves to introduce stochasticity in an otherwise deterministic process. This move from ODE to SDE is not obvious (as acknowledged by reviewer jr5u, and other offline feedback we have received as well). We demonstrate the utility of this empirically in our experiments by studying several special cases (as in Table 1) that allow one to regulate the amount of noise in a sampler and move between deterministic and increasingly stochastic mappings. We also show several qualitative examples of this application in the main paper and appendix. Further, several of the special cases in Table 1 also do not exist in literature, though they follow directly from Corollary 1.2. We would note that while post-hoc one could use FPK equations to shuffle, collapse and rename terms in eq. 6 (Song et al) to arrive at a similar equation as in Corollary 1.2, such a result does not exist in Song et al (2011.13456) and we would argue that our work does that for the reader in a unified way.
> 3. Section 3.3 (Score function from velocity field): While we do not claim novelty for this result in the paper, our proof technique in appendix B.2 is more general and different from prior art to the best of our knowledge. This allowed us to answer reviewer Nc6P’s question about k-Rectified flows. The value of this is similar to how different approaches for solving the same problem are valuable.
> 4. We provide an extensive study of special cases in Table 1 on the toy as well as the large scale ImageNet dataset. These experiments serve to verify Theorem 1 on specialized cases that are similar to the ones studied in literature and apply our results to the application of sampling with a varying level of stochasticity for a given deterministic model. Our experiments further demonstrate that some samplers work better than others, and how our results can be used to get rid of singularities to avoid numerical issues that lead to poor sampling results. We argue that such manipulations are not obvious to a practitioner and our paper provides an easy to use recipe for doing so.

---

### Meta-Review · Area_Chair_yEeH · 2024-12-15

**Metareview:**

This paper presents a stochastic sampling algorithm for deterministic flow (e.g., Rectified flow) models. The main contribution is Theorem 1 which identifies a class of SDEs that share the same marginal distributions. It includes the widely used marginal-equivalent SDE as a special case (Corollary 1.1). The main criticism is that only the algorithm associated with the special case is tested in experiments and used in practice. It is not clearly what the benefits Theorem 1 brings over Corollary 1.1. Reviewer 7s8u also points out that Theorem 1 is a simple exercise in the subject of SDE.

**Additional Comments On Reviewer Discussion:**

The main criticism is that only the algorithm associated with the special case is tested in experiments and used in practice. Reviewer 7s8u also points out that Theorem 1 is a simple exercise in the subject of SDE. The authors respond by emphasizing the novelty of the Theorem 1. The reviewers are not convinced.

---

### Decision · Program_Chairs · 2025-01-22

Reject